# Overcoming Long-Context Limitations of State-Space Models via *Context-Dependent* Sparse Attention

**Zhihao Zhan[1,2,†], Jianan Zhao[1,2,†], Zhaocheng Zhu[1,2], Jian Tang[1,3,4,*]**
[1]Mila - Québec AI Institute, [2]University of Montréal [3]HEC Montréal, [4]CIFAR AI Chair
[†]Equal contribution [*]Correspondence: tangjian@mila.quebec

## Abstract

Efficient long-context modeling remains a critical challenge for natural language processing (NLP), as the time complexity of the predominant Transformer architecture scales quadratically with the sequence length. While state-space models (SSMs) offer alternative sub-quadratic solutions, they struggle to capture long-range dependencies effectively. In this work, we focus on analyzing and improving the long-context modeling capabilities of SSMs. We show that the widely used synthetic task, associative recall, which requires a model to recall a value associated with a single key without context, insufficiently represents the complexities of real-world long-context modeling. To address this limitation, we extend the associative recall to a novel synthetic task, *joint recall*, which requires a model to recall the value associated with a key given in a specified context. Theoretically, we prove that SSMs do not have the expressiveness to solve multi-query joint recall in sub-quadratic time complexity. To resolve this issue, we propose a solution based on integrating SSMs with Context-Dependent Sparse Attention (CDSA), which has the expressiveness to solve multi-query joint recall with sub-quadratic computation. To bridge the gap between theoretical analysis and real-world applications, we propose locality-sensitive Hashing Attention with sparse Key Selection (HAX), which instantiates the theoretical solution and is further tailored to natural language domains. Extensive experiments on both synthetic and real-world long-context benchmarks show that HAX consistently outperforms SSM baselines and SSMs integrated with context-independent sparse attention (CISA). Our code is available at: https://github.com/DeepGraphLearning/HAX.

## 1 Introduction

Long-context modeling is a central challenge in natural language processing (NLP), which underpins a variety of applications, such as document summarization, question answering, and machine translation [30]. Recent advances in large language models (LLMs) have broadened the landscape of long-context modeling, enabling new capabilities such as autonomous agents, retrieval-augmented generation, dialogue systems, and long-context reasoning [22]. This growing demand has spurred intensive research into algorithms that can efficiently and effectively capture long-range dependencies [37].

Currently, the Transformer architecture [38] is the dominant paradigm in sequence modeling. However, its applicability to long sequences is fundamentally constrained by the required computation that grows quadratically with the sequence length. This motivates the research direction for the invention of efficient architectures.

Recently, state-space models (SSMs) [10, 16, 31] have emerged as a potential alternative solution, offering sub-quadratic time complexity as well as comparable performance to Transformers on short-context NLP tasks [14, 4]. However, empirical evidence suggests that SSMs are less effective

39th Conference on Neural Information Processing Systems (NeurIPS 2025).

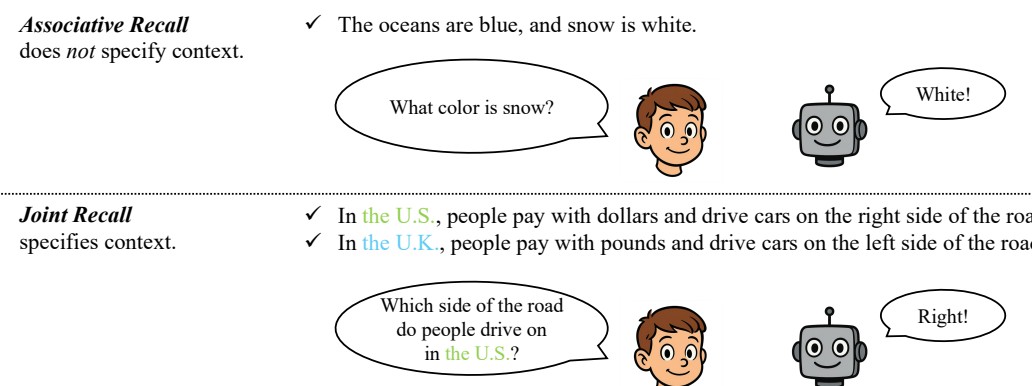

Figure 1: Comparison of joint recall and associative recall. Associative recall does not account for context. *Joint recall* extends associative recall by incorporating *context-dependency* into key-value associations. For example, while associative recall may map "pay with" to either "dollar" or "pound", joint recall allows it to map to "dollar" in the U.S. and "pound" in the U.K., depending on context. This makes joint recall a more realistic and rigorous synthetic task for both theoretical analysis and empirical benchmark for long-context modeling.

than Transformers in capturing long-range dependencies [40]. Furthermore, theoretical analysis by [19] demonstrates that SSMs are much less capable of handling long-context copying, due to the limitations of architecture representation capacity.

In this work, we aim to better understand and improve the long-context modeling abilities of SSMs. We first show that previous studies based on the widely used synthetic task, associative recall [2], might be constrained by its limited capability to simulate natural language in-context dependencies. To be specific, associative recall assumes that each key is uniquely associated with a value, regardless of the surrounding context. However, natural language often defies this assumption: the same key can correspond to different values depending on its context. Consider the example in Figure 1, when asked on which side of the road people drive, the correct answer should depend on the country being referenced. Without specifying whether the context is the US or the UK, the question becomes ambiguous. This example highlights a critical shortcoming of associative recall: it lacks the capacity to simulate context-dependent key-value association, which is very common in natural language.

To address this limitation, we extend the associative recall to a more general synthetic task, joint recall. Unlike associative recall, joint recall requires the model to retrieve a value corresponding to a key conditioned on a specified context. Theoretically, we prove that standard SSMs lack the representational capacity to solve multi-query joint recall under sub-quadratic time complexity.

To overcome this expressiveness bottleneck, we propose to augment SSMs with Context-Dependent Sparse Attention (CDSA), a class of sparse attention with sparse attention patterns that are conditioned on the context representations. Locality-sensitive hashing (LSH) attention [20] exemplifies CDSA, while context-independent sparse attention (CISA) includes sliding window attention, A-shaped attention, and dilated attention [6]. Compared to CISA, CDSA enables dynamic content-dependent routing of information, which is essential to efficiently solve the multi-query joint recall task. We theoretically show that there exists a CDSA which, when integrated with SSMs, enables solving the multi-query joint recall task in sub-quadratic time with respect to sequence length. Moreover, we establish an expressiveness gap between SSMs integrated with CDSA and SSMs integrated with CISA on multi-query joint recall.

Building upon this insight and to bridge the gap between theory and practice, we propose a novel architecture: locality-sensitive Hashing Attention with sparse Key Selection (HAX). HAX improves the expressiveness of LSH attention by incorporating our proposed Key Selection (KS) attention, and is further integrated with state-of-the-art SSMs, Mamba and Mamba2 [14, 4], instantiating the theoretically grounded solution. We validate the effectiveness of HAX through extensive experiments on both synthetic and real-world long-context modeling benchmarks. The experiment results show that HAX consistently outperforms SSM baselines as well as SSMs augmented with CISA. These findings demonstrate that CDSA, when carefully integrated with SSMs, is a critical component in unlocking their potential for long-context modeling.

Our main contributions are summarized as follows:

1. We introduce *joint recall*, a novel synthetic task that extends associative recall to context-dependent key-value association, which offers a new perspective for both theoretical analysis and empirical benchmark for long-context modeling.

2. Through theoretical analysis on *joint recall*, we demonstrate that integrating state-space models (SSMs) with context-dependent sparse attention (CDSA) has the expressiveness to solve multi-query joint recall with sub-quadratic computation.

3. Guided by this theoretical insight, we propose a novel architecture, HAX, based on SSM integrated with CDSA, which consistently outperforms SSMs and SSMs integrated with context-independent sparse attention (CISA) on both synthetic and real-world long-context benchmarks.

## 2 Preliminaries

In this section, we introduce two prominent approaches for efficient architecture design: sparse attention in Sec. 2.1 (exemplified by LSH attention in Sec. 2.2) and SSMs in Sec. 2.3. We also introduce associative recall, a widely-used synthetic benchmark for long-context modeling, in Sec. 2.4.

### 2.1 Sparse Attention

For a sequence of length $l$, we denote the attention scores of auto-regressive sequence modeling as:

$$\mathbf{A} = \text{Softmax}(\mathbf{M} \odot \mathbf{Q}\mathbf{K}^\top) \tag{1}$$

where $\mathbf{M} \in \{0,1\}^{l \times l}$ is the auto-regressive mask, $\mathbf{Q}, \mathbf{K} \in \mathbb{R}^{l \times d}$, $d$ is the hidden dimension. In this work, we define the attention scores for sparse attention as:

$$\mathbf{A} = \text{Softmax}(\mathbf{S} \odot \mathbf{M} \odot \mathbf{Q}\mathbf{K}^\top) \tag{2}$$

with $\mathbf{S} \in \{0,1\}^{l \times l}$ representing the sparse attention pattern. We consider the sparsity constraint:

$$\|\mathbf{S}\|_0 \ll l^2 \tag{3}$$

To ensure per-step computational efficiency, we further tighten this constraint by requiring:

$$\forall i, \|\mathbf{S}_i\|_0 \ll l \tag{4}$$

where $\mathbf{S}_i$ denotes the $i$-th row of $\mathbf{S}$. This implies that each query attends to at most $k$ keys, $k \ll l$.

### 2.2 Locality-Sensitive Hashing Attention

Given that locality-sensitive hashing (LSH) attention represents one of the most effective input-dependent sparse attention for auto-regressive modeling [20], we reformulate a simple algorithm to generate the sparse attention pattern of LSH. This algorithm accepts the query and key matrices $\mathbf{Q}$ and $\mathbf{K}$ as input and outputs a binary sparse attention pattern $\mathbf{S}_{\text{LSH}}$. At each forward pass, $\mathbf{Q}$ and $\mathbf{K}$ are first centralized and normalized to $\tilde{\mathbf{Q}}$ and $\tilde{\mathbf{K}}$, respectively:

$$\tilde{\mathbf{Q}}_i = \text{normalize}(\mathbf{Q}_i - \bar{\mathbf{Q}}_i), \quad \tilde{\mathbf{K}}_i = \text{normalize}(\mathbf{K}_i - \bar{\mathbf{K}}_i) \tag{5}$$

Next, a random projection matrix $\mathbf{H} \overset{\text{i.i.d.}}{\sim} \mathcal{N}(0,1) \in \mathbb{R}^{d \times h}$ is sampled to project the normalized vectors $\tilde{\mathbf{Q}}$ and $\tilde{\mathbf{K}}$ into the hash space. Then, we consider two binning rules which assign each vector $\tilde{\mathbf{Q}}_i$ and $\tilde{\mathbf{K}}_i$ to a hash bin: the argmax binning rule [20, 35] and the sign-bit binning rule [8, 5].

The argmax binning rule assigns each vector to the index of its most aligned column in $\mathbf{H}$:

$$\text{bin}_{Q_i} = \text{argmax}(\tilde{\mathbf{Q}}_i \mathbf{H}), \quad \text{bin}_{K_i} = \text{argmax}(\tilde{\mathbf{K}}_i \mathbf{H}) \tag{6}$$

The sign-bit binning rule constructs a binary hash code by computing the signs of the projected values and interpreting it as a binary number:

$$\text{bin}_{Q_i} = \sum_{j=1}^{h} \mathbf{1}[(\tilde{\mathbf{Q}}_i \mathbf{H})_j > 0] \cdot 2^{h-j}, \quad \text{bin}_{K_i} = \sum_{j=1}^{h} \mathbf{1}[(\tilde{\mathbf{K}}_i \mathbf{H})_j > 0] \cdot 2^{h-j} \tag{7}$$

The argmax binning rule assigns vectors to $h$ bins, while the sign-bit binning rule assigns vectors to $2^h$ bins. We will further discuss the relationship between these two binning strategies in Appendix A. Based on the assigned bins, a preliminary sparse pattern $\tilde{\mathbf{S}}_{\text{LSH}}$ is constructed by allowing each query to attend to all preceding keys within the same bin:

$$\tilde{\mathbf{S}}_{\text{LSH}_{ij}} = \mathbf{1}[\text{bin}_{Q_i} = \text{bin}_{K_j}] \tag{8}$$

Finally, to satisfy the sparsity constraint defined in Eq. 4, a per-bin sliding window mask $\mathbf{M}_{\text{LSH}}$ is applied, so that each query only attends to at most $k$ nearest keys in the same bin:

$$\mathbf{S}_{\text{LSH}} = \mathbf{M}_{\text{LSH}} \odot \tilde{\mathbf{S}}_{\text{LSH}} \tag{9}$$

### 2.3 Generalized State-Space Model

Following the definitions introduced by [19], we formulate generalized state-space models as sequence models defined by an update rule $u : \mathcal{U} \times \mathcal{V} \to \mathcal{U}$ and an output function $r : \mathcal{U} \to \mathcal{V}$, where $\mathcal{V}$ denotes the token vocabulary and $\mathcal{U}$ represents the recurrent state. Let $U_0(\varnothing) \in \mathcal{U}$ denote the initial state. Given an input sequence $v_1, ..., v_n \in \mathcal{V}$, for $i$ in $\{1...n\}$, the state $U_i(v_1, ..., v_i) \in \mathcal{U}$ and its corresponding output $R_i(v_1, ..., v_i) \in \mathcal{V}$ are defined recursively as:

$$U_i(v_1, ..., v_i) = u(U_{i-1}(v_1, ..., v_{i-1}), v_i) \tag{10}$$
$$R_i(v_1, ..., v_i) = r(U_i(v_1, ..., v_i)) \tag{11}$$

### 2.4 Associative Recall

The associative recall task was originally introduced in [2]. [27] found that the LLM performance on this task is strongly correlated with their in-context learning abilities . [1] extended associative recall to the multi-query setting: a model is first given a sequence of associated key-value pairs, and then required to recall each value when queried with the corresponding key. Associative recall has been widely adopted as a synthetic benchmark for long-context modeling [4, 18].

## 3 Joint Recall

We discuss the motivation and formulation of joint recall in Sec. 3.1 and Sec. 3.2, respectively, and finally provide the theoretical results in Sec. 3.3.

### 3.1 Motivation

The motivation behind proposing joint recall is to overcome a key limitation of the setup of associative recall: each key corresponds to a single fixed value. While this setup is well-suited for studying the tasks that emphasize capturing stable lexical patterns, such as sub-word units or fixed multi-word expressions, it falls short in representing the context-sensitive nature of meaning in natural language. Consider the following examples:

- The legislative branch of the U.S. government is called Congress.
- On Monday mornings, Alice studies math.

From a philosophical perspective, definitions are often constructed using genus keys and differentia context. In the first example, the value "Congress"

***Multi-Query Associative Recall*** recalls each value associated with a single key without context:

| Multi-Query Associative Recall | |
|---|---|
| Input | a 5 b 2 c 3 d 1 e 4 \| c ? e ? a ? d ? b ? |
| Output | 3 4 5 1 2 |

***Multi-Query Joint Recall*** recalls each value associated with a key given in a specified context:

| Multi-Query Joint Recall | |
|---|---|
| Input | A a 3 b 2 B b 4 a 1 \| B a ? b ? A b ? a ? |
| Output | 1 4 2 3 |

Figure 2: Comparison between synthetic *multi-query joint recall* and associative recall.

is identified with the genus key "the legislative branch" and the differentia context "of the U.S. government". The second example reflects a more daily scenario, where the value "math" is identifiable only when all contextual elements, "Monday" and "morning", are considered together with the key "Alice". These cases illustrate that accurate semantic interpretation in natural language often requires integrating context and keys, suggesting that models must move beyond the simplistic one-to-one mappings of associative recall to capture the compositional and context-dependent nature of meaning. This observation motivates our introduction of a novel synthetic task, which we refer to as joint recall.

## 3.2 Formulation

Associative recall requires a model to memorize $n_k$ associated key-value pairs. Joint recall generalizes this task: the model is required to memorize an $n_c \times n_k$ table of context-specific key-value associations, in which $n_k$ keys are associated with different values in each of the $n_c$ contexts. Inspired by [1], we also extend joint recall to a multi-query setting, requiring the model to recover the entire table instead of a specific entry in the table.

Figure 2 illustrates multi-query joint recall with $n_c = 2$ and $n_k = 2$. Following the structure of natural language, the sequentialized table input consists of $n_c$ context blocks, each beginning with a context token (e.g. uppercase letter in Figure 2), followed by $n_k$ key-value pairs specific to that context (e.g. lowercase-letter-digit pairs in Figure 2). Then, the model is asked to recall the associated values given each context-key pair, under arbitrary permutations of the context and key ordering.

Appendix C further extends the joint recall formulation to multi-level context: for example, in the sentence "On Monday mornings, Alice studies math.", "Monday" and "morning" are contexts at hierarchical levels. It also provides theoretical analyses grounded in this extended formulation.

## 3.3 Theoretical Analysis

### 3.3.1 Categorization of sparse attention

The sparse attention pattern $\mathbf{S}$ defined in Sec. 2.1 can be categorized as context-dependent or context-independent, depending on whether it is predetermined or dynamically inferred from the context representations. Context-independent sparse attention (CISA) patterns, such as sliding window attention, A-shaped attention, and dilated attention [6], are fixed regardless of context. In contrast, context-dependent sparse attention (CDSA) patterns, exemplified by LSH attention [20], adapt according to the context representation. Appendix Figure 8 provides an illustration of both categories.

### 3.3.2 Limited expressiveness of SSMs

As a corollary of Theorem 2.7 in [19], we demonstrate that solving the multi-query joint recall task imposes a linear growth requirement on the state dimension of SSMs with respect to the number of entries $n$ in the joint recall table, $n = n_c \times n_k$. Let $|\mathcal{U}|$ be the number of distinct representations that the recurrent state space $\mathcal{U}$ can encode, for a state with $b$ bits of capacity, $|\mathcal{U}| = 2^b$. We define the uniform multi-query joint recall distribution as the distribution in which all values are sampled i.i.d. from the uniform distribution over the token vocabulary $\mathcal{V}$. In this setting, we obtain the following:

**Corollary 3.1** (Limited expressiveness of SSMs). *Under the uniform multi-query joint recall distribution, for any $n$, a generalized state-space model defined in Sec. 2.3 incurs an error rate of at least* $1 - \frac{|\mathcal{U}|}{|\mathcal{V}|^n}$.

*Remark 3.1.* To guarantee $\Pr[err] = 0$, it is necessary that the number of representable states satisfies $|\mathcal{U}| \geq |\mathcal{V}|^n$. Taking the logarithm of both sides yields the condition $b \geq n \log |\mathcal{V}|$. This implies that the state-space dimension of the model must grow linearly with the number of entries $n$ in the joint recall table, highlighting a fundamental limitation of the representation capacity of SSMs.

### 3.3.3 Improved expressiveness of SSMs integrated with CDSA

For SSMs integrated with sparse attention, we establish the following results:

**Proposition 3.2** (Improved expressiveness of SSMs integrated with CDSA). *There exists a 2-layer auto-regressive hybrid model consisting of an SSM layer followed by an LSH attention layer, which can solve multi-query joint recall in $O(n \log^2 n)$ time complexity with $O(\log n)$ SSM state dimensions.*

**Proposition 3.3** (Limited expressiveness of SSMs integrated with CISA). *There does not exist a 2-layer auto-regressive hybrid model consisting of an SSM layer followed by a CISA layer, which can solve multi-query joint recall with $o(n^2)$ time complexity, since it requires at least $O(\frac{n}{k})$ SSM state dimensions, $k$ is the maximum number of keys that each query allowed to attend to in the sparse attention module, as defined in Eq. 4.*

*Remark 3.2.* Comparing **Proposition** 3.2 with **Proposition** 3.3, we see a clear representation capacity gap between the SSMs integrated with CDSA and the SSMs integrated with CISA.

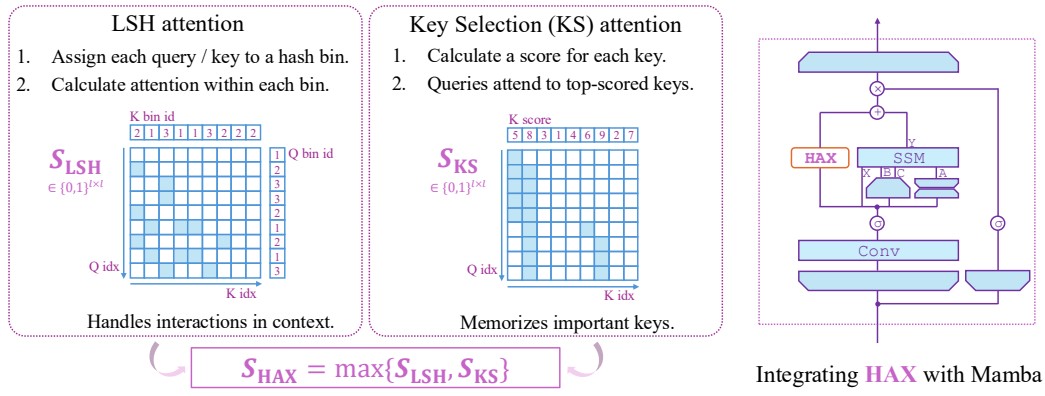

Figure 3: Illustration of the HAX architecture, and Mamba [14] integrated with HAX.

*Remark 3.3.* In practice, with an appropriate constant $k$, integrating CISA with SSMs still provides an advantage: unlike SSMs, which only have access to the last state representation, CISA layers can attend to $k$ different state representations simultaneously, at a cost of $k$ times of computation budget.

Complete proofs are provided in the Appendix B.

## 4 Method

Guided by the theoretical analysis in Sec. 3.3, we propose a new architecture, locality-sensitive Hashing Attention with sparse Key Selection (HAX). HAX improves the expressiveness of LSH attention by incorporating our proposed Key Selection (KS) attention, and is further integrated with state-of-the-art SSMs, serving as an instantiation of context-dependent sparse attention (CDSA) integrated with SSMs, thereby benefiting from the theoretical advantages discussed in Sec. 3.3.

In this section, we first discuss the expressiveness limitations of LSH attention in Sec. 4.1, and then address these limitations by introducing our proposed key selection (KS) attention in Sec. 4.2. Finally, Sec. 4.3 details the architecture of HAX as well as how HAX is integrated with state-of-the-art SSM architectures, Mamba and Mamba2 [14, 4].

### 4.1 Limitation of LSH Attention

In LLMs, certain keys (particularly those at the beginning of a sequence) often receive attention from most queries, forming distinctive "vertical-stripe" attention patterns [39], as illustrated in Appendix Figure 9. These globally attended keys play an essential role in instruction following, where the model is expected to focus its attention on the instruction tokens [23].

Although LSH instantiates CDSA, as discussed in Sec. 2.2, it suffers from a key limitation: difficulty in capturing "vertical-stripe" attention patterns. This arises because in each hashing round, every key is mapped to a single bucket, and attention is constrained to occur only between queries and keys within the same bucket. As a result, when many queries are forced to attend to a limited set of key buckets, those buckets become overloaded, diminishing representation diversity and ultimately degrading attention quality.

### 4.2 Key Selection (KS) Attention

**Goals**. To address the limitation of LSH attention in capturing "vertical-stripe" attention patterns, we propose to augment LSH attention by integrating it with a novel key selection (KS) attention module. This module is designed to satisfy the following desirable properties:

1. "Vertical-stripe" capability: KS attention can express "vertical-stripe" attention patterns.
2. Auto-regressive compatibility: The computation of KS attention for the current token does not depend on future queries or keys.
3. Context-dependent sparsity: KS sparse attention pattern is conditioned on the query and key representations in context and satisfies Eq. 4.

**Modeling.** Taking the query and the key matrices $\mathbf{Q}$ and $\mathbf{K}$ as input, KS attention operates in two phases. The first phase is key scoring, where an scoring module computes an importance score for each key based on the key itself and all previous queries:

$$x_i = f_\theta(\mathbf{K}_i, \mathbf{Q}_{1\ldots i}) \tag{12}$$

The second phase is key selection: each query attends to the $k$ previous highest-scoring keys,

$$\mathbf{S}_{\mathrm{KS}_{ij}} = \mathbf{1}[x_j \in \text{Top-}k\{x_1, ..., x_i\}] \tag{13}$$

With an ideal key scoring module that assigns the highest scores to the globally important keys, KS attention effectively covers $k$ "vertical-stripes" within attention patterns.

For simplicity, we use a multilayer perceptron (MLP) as the key scoring network:

$$f_\theta(\mathbf{K}_i, \mathbf{Q}_{1..i}) \triangleq \mathrm{MLP}\big(\mathbf{K}_i, \mathrm{normalize}\big(\sum_{1 \leq p \leq i} \mathbf{Q}_p\big)\big) \tag{14}$$

**Training.** To train the scoring MLP, at each layer, we randomly sample $k$ candidate keys. Their indices are denoted by $\mathcal{I}$. We compute the reference attention weights via a simple linear attention module, and calculate a pairwise ranking loss between these reference weights and the predicted scores. To be specific, we compute:

$$\mathbf{A}' = \mathbf{Q}\,\mathbf{K}[\mathcal{I}]^\top, \qquad y = \sigma(\mathbf{A}') \odot \mathbf{M}[\mathcal{I}], \tag{15}$$

where $\mathbf{K}[\mathcal{I}]$ is the selected key representations, $\mathbf{M}[\mathcal{I}]$ is the auto-regressive mask restricted to those positions, and $\sigma(\cdot)$ is the sigmoid function. With predicted scores $x \in \mathbb{R}^k$ and target scores $y \in \mathbb{R}^k$, we construct pairwise logits and targets:

$$\mathbf{P}_{ij}(x) = x_i - x_j, \qquad \mathbf{T}_{ij}(y) = \begin{cases} 1 & \text{if } y_i > y_j, \\ 0.5 & \text{if } y_i = y_j, \\ 0 & \text{if } y_i < y_j. \end{cases} \tag{16}$$

and define the ranking loss:

$$\mathcal{L}_{\mathrm{score}}(x, y) = \frac{1}{k^2} \sum_{i,j} \mathrm{BCE}\left(\mathbf{P}_{ij}(x), \mathbf{T}_{ij}(y)\right), \tag{17}$$

where $\mathrm{BCE}(\cdot, \cdot)$ denotes binary cross-entropy. This objective encourages the scoring network to assign higher scores to informative keys that receive higher attention weights. The final training objective sums the ranking loss across layers with the auto-regressive language modeling loss $\mathcal{L}_{\mathrm{LM}}$:

$$\mathcal{L} = \mathcal{L}_{\mathrm{LM}} + \alpha \sum_{\mathrm{layers}} \mathcal{L}_{\mathrm{score}} \tag{18}$$

where $\alpha$ is a scalar hyperparameter that balances the contribution of the ranking loss.

### 4.3 Hybrid Block Design

Finally, we propose locality-sensitive Hashing Attention with sparse Key Selection (HAX), which combines LSH and KS attention patterns:

$$\mathbf{S}_{\mathrm{HAX}} = \max\{\mathbf{S}_{\mathrm{LSH}}, \mathbf{S}_{\mathrm{KS}}\} \in \{0, 1\}^{l \times l} \tag{19}$$

When $\forall i, \|\mathbf{S}_{\mathrm{LSH}_i}\|_0 \leq \frac{k}{2}, \|\mathbf{S}_{\mathrm{KS}_i}\|_0 \leq \frac{k}{2}$, it satisfies

$$\forall i, \|\mathbf{S}_{\mathrm{HAX}_i}\|_0 \leq k \tag{20}$$

Intuitively, LSH and KS attention are complementary, each addressing the other's limitations. LSH attention routes each query to semantically similar keys through randomized hashing, offering flexible, content-based interactions that KS attention alone lacks. In contrast, KS attention introduces broadcast connections to a small set of globally important keys, such as instructions or formatting markers, thereby recovering the "vertical-stripe" patterns that LSH attention struggles to express. While LSH attention promotes diverse contextual representations, mitigating risks of representation collapse, KS attention sharpens focus by allocating attention weights to the most informative positions, enabling stronger long-range control. Importantly, both mechanisms are inherently sparse, so their combination introduces sub-quadratic computational cost.

Figure 3 illustrates Mamba-based and Mamba2-based HAX layer. The proposed hybrid sparse attention layer mitigates the representation capacity limitations of SSMs by coupling them with a parallel sparse attention branch. A parameterized gate rescales the sparse attention output before fusion, which promotes stable optimization.

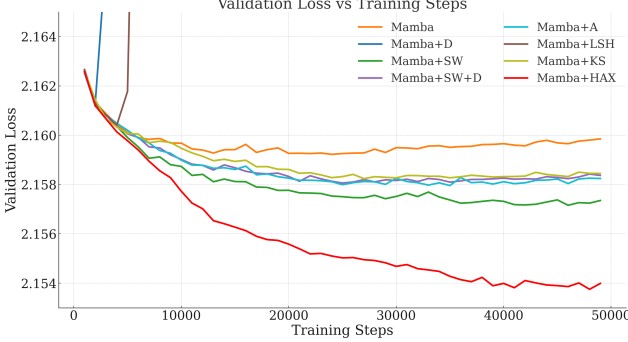

Figure 4: Validation $\mathcal{L}_{LM}$ during continual pre-training on Mamba 130M models. Mamba integrated with HAX is the only architecture that consistently exhibits a decreasing validation loss over the entire training process.

Table 1: Results of multi-query joint recall. Integrating Mamba or Mamba2 with HAX achieves the best performance, which is in **bold**.

|  | Mamba | Mamba2 |
|---|---|---|
| Base | 16.3 | 36.6 |
| +D | 7.8 | 19.6 |
| +SW | 18.7 | 70.6 |
| +SW+D | 16.6 | 48.6 |
| +A | 16.4 | 49.3 |
| +LSH | 11.6 | 13.5 |
| +KS *(ours)* | 36.6 | 60.1 |
| +HAX *(ours)* | **38.0** | **74.3** |
| Base (2×dim) | 34.5 | 59.2 |
| Samba | 6.3 | |

## 5 Experiments

We conduct experiments from two perspectives. First, we empirically verify the theoretical findings presented in Sec. 3 on multi-query joint recall. Then, we demonstrate the effectiveness of the proposed hybrid sparse architecture across synthetic and real-world NLP benchmarks.

### 5.1 Empirical Verification on Multi-Query Joint Recall

**Data.** We construct a multi-query joint recall dataset in which the number of context blocks and the number of keys per context are independently sampled from the range $[5, 16]$, and the size of the vocabulary is fixed at $|\mathcal{V}| = 16$. The dataset consists of 1.4M training examples, along with 14.4K samples each for validation and testing.

**Baselines.** We adopt Mamba [14] and Mamba2 [4] as base architectures and evaluate various hybrid sparse attention models built upon them, as illustrated in Figure 3. These include dilated attention (D), sliding window attention (SW), a combination of sliding window and dilated attention (SW+D) [6], A-shaped attention (A), locality-sensitive hashing (LSH) attention, our proposed key selection (KS) attention, and HAX (a combination of LSH and KS attention). We also consider a Samba baseline.

**Setup.** For fair comparison, we fix the hidden size to $64$ and fix $k = 64$ defined in Eq. 4 (the maximum number of keys each query can attend to) across all hybrid architectures. For variants that integrate multiple sparse attention components, namely SW+D, A (consists of a SW component and a sink attention component, a sink attention only attends to the first tokens in the sequence), and HAX (LSH+KS), we allocate $k = 32$ to each component, in order to maintain a global $k = 64$. We also consider a baseline that doubles the hidden size of SSMs to compensate for the absence of hybrid sparse attention. We fix the number of layers to $2$ for all models. We conduct experiments using 3 different random seeds for each learning rate in {3e-3, 1e-3, 3e-4}, and report the average performance corresponding to the best-performing learning rate. For evaluation, we calculate mean accuracy per-sample and average over the testset.

**Results.** As summarized in Table 1, compared to the Mamba and Mamba2 base architectures, most hybrid sparse attention models improve performance. In particular, our proposed HAX model consistently achieves the best performance, which underscores the effectiveness of our approach.

### 5.2 Continual Pre-training on Natural Language

**Setup.** To evaluate our method on real-world long-context natural language data, we perform continual pre-training based on the publicly released Mamba checkpoints [14]. As in section 5.1, we augment the Mamba architecture with sparse attention, as illustrated in Figure 3. We include SSMs integrated with CISA as baselines, along with ablated variants of our HAX model. We fix the sparsity parameter $k = 128$. For variants that integrate multiple sparse attention components, namely SW+D, A, and HAX (LSH+KS), we set $k = 64$ to each component, in order to maintain a global $k = 128$.

Table 2: Ruler benchmark at 2K context length. We compare different sparse attention integrated with Mamba 790M model, including CISA methods: dilated attention (D), sliding window attention (SW), and their combination (SW+D), and A-shaped attention (A), and CDSA methods: LSH attention, and our proposed key selection (KS) attention and HAX. The best average performance is in **bold**.

| Model | NIAHS1 | NIAHS2 | NIAHS3 | NIAHMK1 | NIAHMK2 | NIAHMK3 | NIAHMV | NIAHMQ | VT | CWE | FWE | QA1 | Average |
|---|---|---|---|---|---|---|---|---|---|---|---|---|---|
| Mamba | 100 | 95.2 | 86.8 | 34.4 | 2.6 | 0 | 32.4 | 28.35 | 2.76 | 1.4 | 100 | 25.2 | 42.43 |
| +D | 0 | 0 | 0 | 0 | 0 | 0 | 0 | 0 | 17.4 | 0 | 100 | 0 | 9.78 |
| +SW | 100 | 93.6 | 83.6 | 36.2 | 3.4 | 0 | 31.05 | 31.45 | 5.24 | 1.26 | 100 | 24.4 | 42.52 |
| +SW+D | 100 | 95 | 82.4 | 34 | 2.6 | 0 | 32.9 | 29.45 | 4.72 | 1.22 | 100 | 24.8 | 42.26 |
| +A | 100 | 93.2 | 81.4 | 34.4 | 2.6 | 0 | 31.7 | 27.1 | 4.4 | 1.26 | 100 | 26.8 | 41.91 |
| +LSH | 0 | 0 | 0 | 0 | 0 | 0 | 0 | 0 | 0.2 | 0.06 | 100 | 0 | 8.36 |
| +KS *(ours)* | 100 | 94.4 | 84.4 | 33.6 | 1.8 | 1.8 | 37.25 | 33.85 | 4.04 | 1 | 100 | 25 | 43.10 |
| +HAX *(ours)* | 100 | 98.8 | 89.6 | 45.4 | 3.8 | 0.2 | 35.5 | 28.2 | 2.04 | 1.64 | 100 | 29.6 | **44.57** |

Table 3: LongBench English tasks. We compare different sparse attention integrated with Mamba 790M model, including CISA and CDSA methods. The best average performance is in **bold**.

| Model | 2WikiMQA | GovReport | HotPotQA | LCC | MultiNews | MultiFQA | MuSiQue | NQA | PsgCnt | PsgRet | Qasper | QMSum | Repobench | SamSum | Trec | TriviaQA | Average |
|---|---|---|---|---|---|---|---|---|---|---|---|---|---|---|---|---|---|
| Mamba | 13.39 | 20.3 | 8.37 | 39.51 | 20.36 | 27.62 | 3.89 | 8.35 | 1.04 | 1.5 | 10.69 | 20.36 | 38.47 | 20.15 | 45 | 37.59 | 19.79 |
| +D | 0 | 0.21 | 0 | 2.87 | 0.26 | 0 | 0 | 0 | 0 | 0 | 0 | 0.72 | 3.46 | 0.7 | 0 | 0 | 0.51 |
| +SW | 13.81 | 20.58 | 9.25 | 38.06 | 20.64 | 27.55 | 5.15 | 8.12 | 1.2 | 1.5 | 10.55 | 19.57 | 38.38 | 22.42 | 43 | 36.55 | 19.77 |
| +SW+D | 13.6 | 20.39 | 8.55 | 39 | 20.74 | 27.55 | 4.33 | 7.98 | 0.83 | 1.56 | 10.17 | 19.73 | 38.84 | 22.49 | 43 | 37.02 | 19.74 |
| +A | 13.63 | 20.77 | 8.55 | 38.59 | 20.2 | 27.31 | 4.56 | 7.66 | 0.27 | 1.74 | 11.32 | 20.09 | 38.5 | 20.73 | 44.5 | 37.16 | 19.72 |
| +LSH | 1.66 | 1.49 | 1.5 | 7.74 | 0.98 | 2.32 | 0.96 | 0.87 | 0.1 | 0 | 2.3 | 5.05 | 7.13 | 3.89 | 0 | 0.9 | 2.31 |
| +KS *(ours)* | 12.93 | 20.93 | 9.34 | 38.22 | 20.83 | 26.56 | 4.34 | 7.46 | 0.27 | 1.67 | 11.65 | 19.35 | 38.17 | 22.97 | 45 | 37.52 | 19.83 |
| +HAX *(ours)* | 12.73 | 20.64 | 8.79 | 39.3 | 21.79 | 28.31 | 3.95 | 7.64 | 0.83 | 2 | 11.94 | 19.63 | 39.07 | 21.04 | 43 | 38.47 | **19.95** |

Table 4: Performance of extrapolation from 2K to 4K context length on Ruler benchmark. We compare different sparse attention integrated with Mamba 790M model, including CISA and CDSA methods as in Table 2. The best average performance is in **bold**.

| Model | NIAHS1 | NIAHS2 | NIAHS3 | NIAHMK1 | NIAHMK2 | NIAHMK3 | NIAHMV | NIAHMQ | VT | CWE | FWE | QA1 | QA2 | Average |
|---|---|---|---|---|---|---|---|---|---|---|---|---|---|---|
| Mamba | 100 | 19.2 | 9.2 | 5.2 | 0.6 | 0 | 0.65 | 1.85 | 5.72 | 4.3 | 100 | 26.2 | 78.6 | 27.04 |
| +D | 0 | 0 | 0 | 0 | 0 | 0 | 0 | 0 | 65.2 | 0 | 100 | 0 | 73 | 18.32 |
| +SW | 100 | 18.2 | 5 | 5.6 | 0.2 | 0.2 | 1.75 | 2.1 | 4.4 | 4.78 | 100 | 22.4 | 78.8 | 26.42 |
| +SW+D | 100 | 20.2 | 8.2 | 6.2 | 0.4 | 0.2 | 1.45 | 1.25 | 4.36 | 4.32 | 100 | 24.6 | 79 | 26.94 |
| +A | 100 | 19.8 | 7.2 | 6.6 | 1.2 | 0.2 | 1.3 | 0.8 | 4.36 | 3.24 | 100 | 24.2 | 79 | 26.76 |
| +LSH | 0 | 0 | 0 | 0 | 0 | 0 | 0 | 0 | 0 | 0.04 | 100 | 0.2 | 73 | 13.33 |
| +KS *(ours)* | 99.8 | 20.2 | 7.8 | 8.2 | 1 | 0.2 | 1.2 | 1.85 | 9.32 | 2.86 | 100 | 22.6 | 78.6 | 27.20 |
| +HAX *(ours)* | 100 | 23 | 11 | 9 | 0.8 | 0 | 3.4 | 2.75 | 6.36 | 2.54 | 100 | 29 | 78.8 | **28.20** |

Overall, our HAX adds only about 1% additional parameters to the released Mamba architecture. All models are continually pre-trained for 50K steps with a context length of 2K.

**Validation Loss during Continual Pre-Training.** Figure 4 shows the validation loss $\mathcal{L}_{LM}$ of 130M model continual pre-training on The Pile [11]. As observed, the Mamba base architecture and all baseline variants exhibit either training instability or plateau early in the training process. In contrast, our proposed HAX model is the only architecture that shows a consistent decline in validation loss throughout the training process, indicating improved stability and sustained learning.

**Ruler and LongBench Evaluation.** Tables 2 and 3 present the downstream performance for the Mamba 790M model after continual pre-training for 50K steps on TxT-360 [36] followed by instruction fine-tuning for 10K steps on UltraChat [7] following Mamba-Chat [24]. Ruler [18] is a synthetic long-context NLP benchmark designed to assess model performance on tasks including retrieval, multi-hop reasoning, aggregation, and question answering. LongBench [3] comprises real-world NLP long-context tasks, including question answering, summarization, few-shot learning, retrieval, aggregation, and code completion. The evaluation results, summarized in Tables 2 and 3, show that among all hybrid sparse attention variants, our proposed HAX model is the only one that outperforms the Mamba baseline by a significant margin on average.

**Extrapolation Evaluation.** We additionally evaluate the Mamba 790M models on the Ruler [18] benchmark at a 4K context length, despite being continually pre-trained and instruction fine-tuned solely with a 2K context length. As summarized in Table 4, our proposed HAX model consistently outperforms all baselines, underscoring its robustness for context length extrapolation.

# 6 Related Work

## 6.1 State-Space Models

State-space models (SSMs) originated in control theory, exemplified by damped mass-spring systems [29]. HiPPO [15] was one of the first efforts to adapt SSMs for machine learning applications. LSSL [17] unified convolutional neural networks (CNNs), recurrent neural networks (RNNs), and ordinary differential equations (ODEs) under the SSM framework, enabling their implementation within deep neural networks. H3 [10] integrated SSM layers with short convolutional filters to enhance sequence modeling. Mamba [14] advanced this line of work by making all SSM parameters input-dependent, which significantly increases the representation capacity of SSMs. Mamba2 [4] further improved the architecture and established connections between SSMs and Transformer attention.

## 6.2 Analysis on State-Space Models

Recent empirical studies have shown that SSM long-context modeling performance often lags behind that of Transformer architectures [40]. In theory, [19] demonstrated that even solving simple tasks like copying requires SSM state dimensions to grow linearly with the sequence length. Furthermore, [25] established that the expressiveness of linear or diagonal SSMs is bounded by the complexity class $TC^0$. [34] further showed that SSMs and Transformers capture overlapping yet distinct subsets of $TC^0$, providing a theoretical basis for developing hybrid models that combine the strengths of both architectures.

## 6.3 Context-Dependent Sparse Attention

Locality-sensitive hashing (LSH) attention [20] stands as one of the most widely adopted forms of context-dependent sparse attention. Follow-up LSH attention variants can be broadly categorized by their binning rules: the argmax binning rule is adopted in [20, 35], while the sign-bit binning rule is used in [8, 5]. [28] introduces a Triton kernel that accelerates hash-based sparse attention. Recently, [43] proposed native sparse attention, a new context-dependent sparse attention that outperforms Transformers training from scratch, highlighting its potential for efficient long-context modeling.

## 6.4 Hybrid Architectures

Several works have explored architectures that mix a large proportion of SSM layers with a small number of full attention layers, and have reported performance surpassing that of standard Transformers [40]. The effectiveness of such hybrid architectures has been further validated at billion-parameter scale [21]. In parallel, researchers have also investigated the design of hybrid sparse attention models [33, 32, 9, 26, 44], which offer sub-quadratic computational complexity, providing a promising direction for efficient long-context modeling.

# 7 Conclusion

In this work, we introduce *joint recall*, a novel synthetic task that generalizes associative recall to context-dependent key-value retrieval. Theoretically, we show that both SSMs and SSMs integrated with context-independent sparse attention (CISA) could not solve multi-query joint recall within sub-quadratic time complexity, while integrating SSMs with context-dependent sparse attention (CDSA) overcomes this limitation. Guided by this insight, we propose to integrate state-of-the-art SSMs with a novel CDSA, locality-sensitive Hashing Attention with sparse Key Selection (HAX). Experiment results confirm that HAX achieves improved training stability and consistent performance gains across synthetic and real-world long-context NLP benchmarks. The *joint recall* task therefore offers a unified theoretical lens and empirical yardstick for long-context modeling, while HAX demonstrates the power of theory-driven architecture design. These results highlight the importance of aligning model design with expressiveness improvements, and demonstrate that combining efficient sequence models with CDSA is a promising direction for scalable long-context modeling.

## Acknowledge

We would like to thank Jingyang Yuan, Wanru Zhao, Meng Qu, Zewen Chi for their insightful discussion on language model pre-training. We also thank the anonymous reviewers, AC, SAC, and PC for their time, thoughtful feedback, and constructive engagement throughout the review process. We also acknowledge funding from the Canada CIFAR AI Chair Program and the Intel-Mila partnership program. The computation resource of this project is supported by Mila and the Digital Research Alliance of Canada.

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

Figure 5: Input and output components of the auto-regressive multi-query joint recall task. The input sequence is further divided into an information component and an inquiry component.

## A  Relationship Between the Argmax and Sign-Bit LSH Binning Rules

In this section, we show how the sign-bit LSH binning rule (Eq. 7) can be interpreted as an argmax LSH binning rule (Eq. 6) applied to an expanded projection matrix with $2^h$ columns. We first construct the expanded matrix, and then prove the equivalence.

**Expanding the projection matrix.** Let the original random projection be $\mathbf{H} = [\mathbf{H}_1, ..., \mathbf{H}_h] \in \mathbb{R}^{d \times h}, \mathbf{H}_j \overset{\text{i.i.d.}}{\sim} \mathcal{N}(0, 1)$. Define a codebook of $2^h$ signed prototypes

$$\mathcal{B} = \left\{ \mathbf{B_b} = \sum_{j=1}^{h} b_j \mathbf{H}_j \,\middle|\, \mathbf{b} = (b_1, \ldots, b_h) \in \{-1, +1\}^h \right\} \subset \mathbb{R}^d. \tag{21}$$

Stacking all $\mathbf{B_b}$ as columns yields the implicit matrix $\tilde{\mathbf{H}} \in \mathbb{R}^{d \times 2^h}$.

**Equivalence of the two binning rules.** For a normalized query vector $\tilde{\mathbf{Q}}_i$, we define its sign projection as $\mathbf{s} = \text{sign}(\tilde{\mathbf{Q}}_i^\top \mathbf{H}) \in \{-1, +1\}^h$. The inner product of $\tilde{\mathbf{Q}}_i$ and a prototype $\mathbf{B_b} \in \mathcal{B}$ is

$$\langle \tilde{\mathbf{Q}}_i, \mathbf{B_b} \rangle = \sum_{j=1}^{h} b_j \langle \tilde{\mathbf{Q}}_i, \mathbf{H}_j \rangle. \tag{22}$$

Because every term with $b_j \neq s_j$ flips the sign of the positive quantity $|\langle \tilde{\mathbf{Q}}_i, \mathbf{H}_j \rangle|$, we have the strict inequality $\langle \tilde{\mathbf{Q}}_i, \mathbf{B_s} \rangle > \langle \tilde{\mathbf{Q}}_i, \mathbf{B_b} \rangle$ for all $\mathbf{b} \neq \mathbf{s}$. Hence

$$\text{argmax}_{\mathbf{b} \in \{-1, +1\}^h} \langle \tilde{\mathbf{Q}}_i, \mathbf{B_b} \rangle = \mathbf{s} = \text{bin}_{Q_i}^{\text{(sign)}}, \tag{23}$$

The sign-bit assignment is exactly the argmax rule applied to $\tilde{\mathbf{H}}$. An identical argument holds for keys $\tilde{\mathbf{K}}_j$. Thus, the sign-bit method equals the argmax method with $2^h$ (expanded) columns.

## B  Theoretical Proof

Multi-query joint recall requires models to recall an $n_c \times n_k$ table of context-specific key-value associations, in which $n_k$ keys are associated with different values in each of the $n_c$ contexts, with $n = n_c \times n_k$ being the total number of entries. For clarity, we introduce some additional notations for multi-query joint recall in the auto-regressive setting, as illustrated in Appendix Figure 5. The input sequence is divided into an information component and an inquiry component. The information component provides the context-specific key-value associations. The inquiry component permutes the order of context and keys in the information component, and the model is required to predict the corresponding values given each key under every specified context.

### B.1  Proof of Corollary 3.1

**Corollary 3.1** (Limited expressiveness of SSMs). *Under the uniform multi-query joint recall distribution, for any $n$, a generalized state-space model defined in Sec. 2.3 incurs an error rate of at least $1 - \frac{|\mathcal{U}|}{|\mathcal{V}|^n}$.*

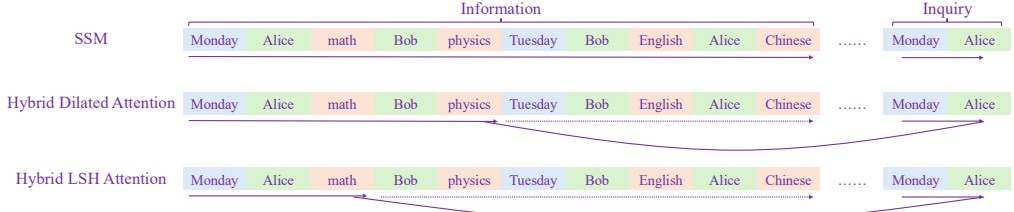

Figure 6: Comparison between SSM, hybrid dilated attention model and hybrid locality-sensitive hashing (LSH) attention model on joint recall. By selectively bypassing irrelevant context, sparse attention alleviates memory overload in SSM layers and enhances the hybrid model's capability to retrieve relevant information.

The intuition behind the proof of Corollary 3.1 is straightforward: the number of possible joint recall data instances that a model can accurately represent is fundamentally limited by the capacity of its recurrent state. Since all information from the input sequence must be encoded into a fixed recurrent state following the context during processing, the total number of distinguishable output is bounded by the representation capacity of the state space $|\mathcal{U}|$. Consequently, if the size of the output space $|\mathcal{V}|^n$ exceeds $|\mathcal{U}|$, the model inevitably incurs non-negligible error.

As a direct consequence of Theorem 2.7 in [19], we adopt their proof strategy. We reformulate Lemma D.1 in [19] as the following Lemma B.1. Let $m$ denote the index of the last token in the information component. Then, for any fixed permutation $\mathcal{P}$ of the context and keys in the inquiry component, the following Lemma B.1 holds:

**Lemma B.1.** *Let $\mathcal{M}$ be a fixed-state generalized SSM that maps the joint recall input space $\mathcal{X}$ to the output space $\mathcal{V}^n$ under any fixed permutation $\mathcal{P}$ of the context and keys in the inquiry component. Then there exists a function $G : \mathcal{U} \to \mathcal{V}^n$ such that for all inputs $\mathbf{x} \in \mathcal{X}$, the model output satisfies $\mathcal{M}(\mathbf{x}) = G(U_m(\mathbf{x}))$, $U_m$ is defined in Eq.10.*

Following [19], we bound the error of the model by comparing the number of possible model states to the number of distinct input instances.

*Proof.*

$$1 - \Pr[err] = \Pr[\mathcal{M}(\mathbf{x}) = \mathbf{y}|\mathbf{y} \in \mathcal{V}^n] \tag{24}$$

$$= \frac{1}{|\mathcal{V}|^n} \sum_{\mathbf{y} \in \mathcal{V}^n} \mathbf{1}[\mathcal{M}(\mathbf{x}) = \mathbf{y}] \tag{25}$$

$$= \frac{1}{|\mathcal{V}|^n} \sum_{\mathbf{y} \in \mathcal{V}^n} \sum_{\mathbf{u} \in \mathcal{U}} \mathbf{1}[G(\mathbf{u}) = \mathbf{y}] \cdot \mathbf{1}[U_m(\mathbf{x}) = \mathbf{u}] \tag{26}$$

$$\leq \frac{1}{|\mathcal{V}|^n} \sum_{\mathbf{u} \in \mathcal{U}} \mathbf{1}[U_m(\mathbf{x}) = \mathbf{u}] \tag{27}$$

$$\leq \frac{|\mathcal{U}|}{|\mathcal{V}|^n} \tag{28}$$

$\square$

To guarantee $\Pr[err] = 0$, it is necessary that the number of representable states satisfies $|\mathcal{U}| \geq |\mathcal{V}|^n$. Taking the logarithm of both sides yields the condition $b \geq n \log |\mathcal{V}|$. This implies that the state-space dimension of the model must grow linearly with the number of entries $n$ in the joint recall table, highlighting a fundamental limitation of the representation capacity of SSMs.

In contrast, as illustrated in Figure 6, hybrid sparse attention models mitigate this limitation by enabling information to propagate through multiple parallel paths, thereby alleviating the bottleneck imposed by sequential state updates.

## B.2 Proof of Proposition 3.2

**Proposition 3.2** (Improved expressiveness of SSMs integrated with CDSA). *There exists a 2-layer auto-regressive hybrid model consisting of an SSM layer followed by an LSH attention layer, which can solve multi-query joint recall in $O(n \log^2 n)$ time complexity with $O(\log n)$ SSM state dimensions.*

*Proof.* We prove by construction. In the first layer, we expect the SSM state concatenates each value token representation with its associated key token representation and context token representation. To be specific, we expect the SSM state representation at each value token to be

$$[\mathbf{c}, \mathbf{k}, \mathbf{v}, is\_v] \in \mathcal{U}$$

where $\mathbf{c}$ is the representation of the current associated context token, $\mathbf{k}$ is the representation of the current associated key token, and $\mathbf{v}$ is the representation of the nearest value token. $is\_v$ is a sign indicator ($-1$ or $1$) that specifies whether the current token is a value token.

To achieve this, we first construct each vector $\mathbf{c}$, $\mathbf{k}$ and $\mathbf{v}$ be a distinct $b$-dimensional vector with unit norm without zero entries, i.e., $\|\mathbf{c}\|_2 = 1$, $\|\mathbf{k}\|_2 = 1$, $\|\mathbf{v}\|_2 = 1$, $\forall j, \mathbf{c}_j \neq 0, \mathbf{k}_j \neq 0, \mathbf{v}_j \neq 0$. Since the number of distinct vectors that can be drawn from the unit sphere grows exponentially with dimensionality, $O(\log n)$ embedding dimensions are sufficient to ensure that all representations are distinguishable. Then we define an embedding space in which context and value tokens are mapped to structured representations. Specifically, a context token is embedded as

$$\mathbf{e} = [\mathbf{c}, \mathbf{0}, \mathbf{0}, -1]$$

where $\mathbf{c}$ is the constructed representation of this context token on the unit sphere, and the final coordinate is set to $-1$ to indicate that the current token is not a value. Similarly, a key token is embedded as

$$\mathbf{e} = [\mathbf{0}, \mathbf{k}, \mathbf{0}, -1]$$

and a value token is embedded as

$$\mathbf{e} = [\mathbf{0}, \mathbf{0}, \mathbf{v}, 1]$$

where $\mathbf{k}$ and $\mathbf{v}$ are the key and value representations from the unit sphere, respectively, and the final coordinate is set to $1$ only when the current token is a value token. Following Eq. 10, we define the update rule of the generalized SSM as follows:

$$U_i = u(U_{i-1}, \mathbf{e}) = U_{i-1} \odot \mathbf{1}[\mathbf{e}_j = 0] + \mathbf{e} \odot \mathbf{1}[\mathbf{e}_j \neq 0] \tag{29}$$
$$R_i = r(U_i) = U_i \tag{30}$$

where $\mathbf{e}$ is the current input embedding and $\mathbf{e}_j$ refers to its $j$-th dimension. The update rule operates as a conditional overwrite: if a position does not carry information (i.e., the corresponding dimension in $\mathbf{e}$ is $\mathbf{0}$), the previous state is preserved; otherwise, it is updated with the current embedding. Following this update rule, the SSM state at each value token in the information component takes the form

$$[\mathbf{c}, \mathbf{k}, \mathbf{v}, 1]$$

while the SSM state at each key token takes the form

$$[\mathbf{c}, \mathbf{k}, ?, -1]$$

In the second layer, LSH attention operates on the SSM state $[\mathbf{c}, \mathbf{k}, \mathbf{v}, is\_v] \in \mathcal{U}$, using $[\mathbf{c}, \mathbf{k}, \mathbf{0}, is\_v]$ as the LSH attention key representation, $[\mathbf{c}, \mathbf{k}, \mathbf{0}, 1]$ as the LSH attention query representation, and $[\mathbf{0}, \mathbf{0}, \mathbf{v}, 1]$ as the LSH attention value representation. This design ensures that value tokens in the information component and key tokens in the inquiry component that share the same context and key (i.e., identical $\mathbf{c}$ and $\mathbf{k}$ representations in the SSM state) will always be assigned to the same hash bin. With a sufficient number of, e.g. $O(n)$ hash bins, which can be efficiently constructed using sign-bit binning rule with $O(\log n)$ random projection vectors, values associated with each key in every specified context are reliably retrievable by LSH attention. This step is the bottleneck of computation with $O(n \log^2 n)$ time complexity. □

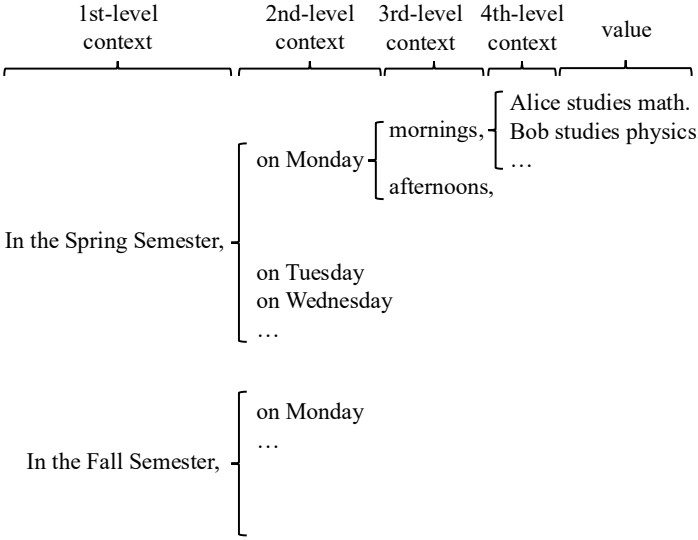

Figure 7: An example of multi-level context in natural language.

### B.3 Proof of Proposition 3.3

**Proposition 3.3** (Limited expressiveness of SSMs integrated with CISA). *There does not exist a 2-layer auto-regressive hybrid model consisting of an SSM layer followed by a CISA layer, which can solve multi-query joint recall with $o(n^2)$ time complexity, since it requires at least $O(\frac{n}{k})$ SSM state dimensions, $k$ is the maximum number of keys that each query allowed to attend to in the sparse attention module, as defined in Eq. 4.*

*Proof.* Consider a key given in the inquiry component of the auto-regressive joint recall task. The model is required to output the associated value token when this key token is provided as input. Taking the query representation from this key token, the sparse attention can attend to at most $k$ key representations from previous tokens, where the key representations are calculated based on the SSM state representations of the first layer. To solve the joint recall task, these $k$ key representations being attended must collectively encode the full information component. Since the full information component length is $O(n)$, by Corollary 3.1, $k$ state representations of the generalized SSM in the first layer must use at least $O(\frac{n}{k})$ dimensions to collectively store the information component. Thus, the per-key computational cost required by the second-layer sparse attention is $O(k \cdot \frac{n}{k}) = O(n)$, and therefore the total time complexity is $O(n^2)$. □

Comparing **Proposition** 3.2 with **Proposition** 3.3, we see a clear representation capacity gap between the SSMs integrated with CDSA and the SSMs integrated with CISA. In practice, however, with an appropriate constant $k$, integrating CISA with SSMs still provides an advantage: unlike SSMs, which only have access to the last state representation, CISA layers can attend to $k$ different state representations simultaneously, at a cost of $k$ times of computation budget.

## C  Extending Joint Recall to Multi-level Context

As illustrated in Figure 7, in many cases, natural language contexts exhibit hierarchical dependencies. This motivates us to extend joint recall to the multi-level context setting, in which we regard the keys as the last level of context.

### C.1  Formulation

Given $w$ different levels of context vocabulary $\mathcal{C}_1, \mathcal{C}_2, ..., \mathcal{C}_w$ and token vocabulary $\mathcal{V}$, multi-level context joint recall requires a model to recover the mapping $\mathcal{C}_1 \times \mathcal{C}_2 \times ... \times \mathcal{C}_w \to \mathcal{V}$. The context of multi-level context joint recall is hierarchically structured analogously to natural languages. It is

divided into $|\mathcal{C}_1|$ first-level blocks, where each first-level block begins with a token from the first-level context vocabulary $\mathcal{C}_1$. Each first-level block is further divided into sub-blocks beginning with a second-level context token from $\mathcal{C}_2$, and this recursive sub-division continues up to the $w$-th level. The last-level block consists of a $w$-th level context token followed by a value token from $\mathcal{V}$. Note that associative recall is a special case of multi-level joint recall with $w = 1$, and joint recall is a special case of multi-level joint recall with $w = 2$. We similarly define multi-query multi-level context joint recall, where the model is required to recall all $n = |\mathcal{C}_1| \times |\mathcal{C}_2| \times ... \times |\mathcal{C}_w|$ entries in the full context-value supertable.

## C.2 Expressiveness of SSMs Integrated with CDSA on Multi-Level Context Joint Recall

On multi-query multi-level context joint recall, both Corollary 3.1 and Proposition 3.3 continue to hold under the same assumptions. We now extend Proposition 3.2 to the following Proposition C.1, which demonstrates that SSMs integrated with CDSA remain expressive even in the presence of $w$ levels of hierarchical contexts.

**Proposition C.1** (Expressiveness of SSMs integrated with CDSA on multi-level context joint recall)**.** *There exists a 2-layer auto-regressive hybrid model consisting of an SSM layer followed by an LSH attention layer, which can solve multi-query multi-level context joint recall in $O(wn \log^2 n)$ time complexity with $O(w \log n)$ SSM state dimensions.*

*Proof.* Similar to Proposition 3.2, we prove by construction. In the first layer, we hope the SSM state to consist of the context and value representations

$$[\mathbf{z}_1, \mathbf{z}_2, ..., \mathbf{z}_w, \mathbf{v}, is\_v] \in \mathcal{U}$$

where $\mathbf{z}_i$ denotes the representation of the nearest $i$-th level context token, $\mathbf{v}$ represents the nearest value token, and $is\_v$ is a sign indicator ($-1$ or $1$) that specifies if the current token is a value token.

To achieve this, we similarly construct each vector $\mathbf{z}_i$ and $\mathbf{v}$ to be a distinct $b$-dimensional vector with unit norm without zero entries, i.e., $\|\mathbf{z}_i\|_2 = 1$, $\|\mathbf{v}\|_2 = 1$, $\forall j, \mathbf{z}_{ij} \neq 0, \mathbf{v}_j \neq 0$. Since the number of distinct vectors that can be drawn from the unit sphere grows exponentially with dimensionality, $O(\log n)$ embedding dimensions are sufficient to ensure that all representations are distinguishable. Consequently, the total size of the SSM state is $O(w \log n)$.

Then we define an embedding space in which context and value tokens are mapped to structured representations. Specifically, a $i$-th level context token is embedded as

$$\mathbf{e} = [\mathbf{0}, ..., \mathbf{0}, \mathbf{z}_i, \mathbf{0}, ..., -1]$$

where $\mathbf{z}_i$ is the context token representation, and the final coordinate is set to $-1$ to indicate that the token is not a value. Similarly, a value token is embedded as

$$\mathbf{e} = [\mathbf{0}, \mathbf{0}, ..., \mathbf{0}, \mathbf{v}, 1]$$

where $\mathbf{v}$ is the value token representation and the final coordinate is set to $1$ to mark it as a value token. We keep the update rule of the generalized SSM as in Eq. 29:

$$U_i = u(U_{i-1}, \mathbf{e}) = U_{i-1} \odot \mathbf{1}[\mathbf{e}_j = 0] + \mathbf{e} \odot \mathbf{1}[\mathbf{e}_j \neq 0] \tag{31}$$

$$R_i = r(U_i) = U_i \tag{32}$$

which operates as a conditional overwrite. Following this update rule, the SSM state at each value token takes the form

$$[\mathbf{z}_1, \mathbf{z}_2, ..., \mathbf{z}_w, \mathbf{v}, 1]$$

where $\mathbf{z}_1, \mathbf{z}_2, ..., \mathbf{z}_w$ are the $w$ levels of context representations of the current token, $\mathbf{v}$ is the value representation, and the final dimension is set to $1$ to indicate that the current token is a value token.

In the second layer, LSH attention operates on the SSM state $[\mathbf{z}_1, \mathbf{z}_2, ..., \mathbf{z}_w, \mathbf{v}, is\_v]$, using $[\mathbf{z}_1, \mathbf{z}_2, ..., \mathbf{z}_w, \mathbf{0}, is\_v]$ as the LSH attention key representation, $[\mathbf{z}_1, \mathbf{z}_2, ..., \mathbf{z}_w, \mathbf{0}, 1]$ as the LSH attention query representation, and $[\mathbf{0}, \mathbf{0}, ..., \mathbf{v}, 1]$ as the LSH attention value representation. This design ensures that tokens that share the same context (i.e., identical $[\mathbf{z}_1, \mathbf{z}_2, ..., \mathbf{z}_w]$ in the SSM state) will always be assigned to the same hash bin. With a sufficient number of, e.g. $O(n)$ hash bins, which can be efficiently constructed using sign-bit binning rule with $O(\log n)$ random projection

vectors, value representations associated with identical key combinations in the context are reliably retrievable by LSH attention. This step is the bottleneck of computation with $O(wn \log^2 n)$ time complexity. □

This construction establishes that a 2-layer hybrid model consisting of a generalized SSM followed by LSH attention can solve multi-query multi-level joint recall efficiently, with sub-quadratic time complexity and sub-linear state complexity with respect to the input sequence length.

# D   Experiment Details

## D.1   Empirical Verification on Joint Recall

For all models, we fix the number of layers to 2, set the hidden size to 64, and use $k = 64$. For variants that integrate multiple sparse attention components, namely SW+D, A (consists of a SW component and a sink attention component, a sink attention always attends to the first $k$ tokens in the sequence only), and HAX (LSH+KS), we allocate $k = 32$ to each component, in order to maintain a global $k = 64$. For both LSH and HAX (LSH+KS), we adopt the sign-bit binning strategy (Eq. 7) with $h = 8$, and refresh the random hashing matrix at each training step. To account for the absence of sparse attention mechanisms, we additionally evaluate Mamba and Mamba2 baselines with their hidden size doubled to 128, ensuring a fair comparison. Additionally, we include a Samba baseline, consisting of 2 Mamba layers and 2 sliding window attention layers. For Samba, the hidden size and sliding window width ($k$) are both set to 64. We use AdamW optimizer. All models are trained for 400,000 steps with a batch size of 64. Our implementation is based on Flash-Linear-Attention [42].

## D.2   Continual Pre-training on Natural Language

For all experiments, we fix the sparsity parameter $k = 128$. For variants that integrate multiple sparse attention components, namely SW+D, A, and HAX (LSH+KS), we set $k = 64$ to each component, in order to maintain a global $k = 128$. For both LSH and HAX (LSH+KS), we adopt the argmax binning strategy (Eq. 6) with $h = k$. We resample the random hashing matrix at each training step and fix a random hashing matrix during evaluation.

To enhance the long-context modeling capability, we filter samples to retain only those with tokenized lengths of at least 2,048 tokens. At the beginning of continual pre-training, the $\mathbf{K}$ and $\mathbf{Q}$ projection weights are initialized using the parameters of the $\mathbf{B}$ and $\mathbf{C}$ projections from Mamba, respectively, based on state-space duality [4]. During continual pretraining, we use a cosine schedule with a maximum learning rate of 3e-4 for 130M models and 1.5e-4 for 790M models, and a minimum learning rate of 1e-5. A warm-up phase of 200 steps with a learning rate of 0 followed by 800 steps of linear warm-up precedes the cosine schedule. For instruction tuning, we also apply a 200-step 0 learning rate phase followed by 800 steps of linear warm-up, after which the learning rate remains constant at 3e-6. For both continual pre-training and instruction-tuning, we use AdamW optimizer and train with a global batch size of 64 and a context length of 2K.

# E   Additional Experiments

## E.1   Short-context Modeling Benchmark

We follow Mamba [14] to evaluate the zero-shot short context modeling performance of the continually pre-trained models on the LM evaluation harness benchmark from EleutherAI [12]. Our results in Table 5 show that continual pre-training on long sequences will not lead to a significant performance drop on short context benchmarks, where the Mamba w/o continual pre-training results are copied from the Mamba paper [14].

Table 5: LM evaluation harness benchmark for continually pre-trained Mamba models. We compare different sparse attention integrated with Mamba, including CISA and CDSA methods as in Table 2.

|  | LambdaPPL | WinoGrande | PIQA | LambdaAcc | HellaSwag | ARC-E | ARC-C | AverageAcc |
|---|---|---|---|---|---|---|---|---|
| Mamba | 15.58 | 51.8 | 63.8 | 44.3 | 35.3 | 47.8 | 24.4 | 44.6 |
| +D | 15.75 | 52.8 | 64.4 | 43.9 | 35.3 | 47.7 | 24.1 | 44.7 |
| +SW | 15.47 | 52.8 | 63.8 | 44.3 | 35.2 | 47.9 | 24.7 | 44.8 |
| +SW+D | 15.35 | 53.0 | 64.0 | 44.7 | 35.2 | 47.6 | 24.4 | 44.8 |
| +A | 15.65 | 53.0 | 64.0 | 44.2 | 35.3 | 47.7 | 24.7 | 44.8 |
| +LSH | 15.73 | 52.6 | 64.3 | 43.8 | 35.3 | 47.7 | 24.0 | 44.6 |
| +KS *(ours)* | 15.86 | 52.4 | 63.7 | 44.1 | 35.2 | 47.9 | 24.5 | 44.6 |
| +HAX *(ours)* | 15.62 | 52.5 | 63.9 | 44.3 | 35.3 | 47.9 | 24.5 | 44.7 |
| w/o training | 16.07 | 51.9 | 64.5 | 44.3 | 35.3 | 48.0 | 24.3 | 44.7 |

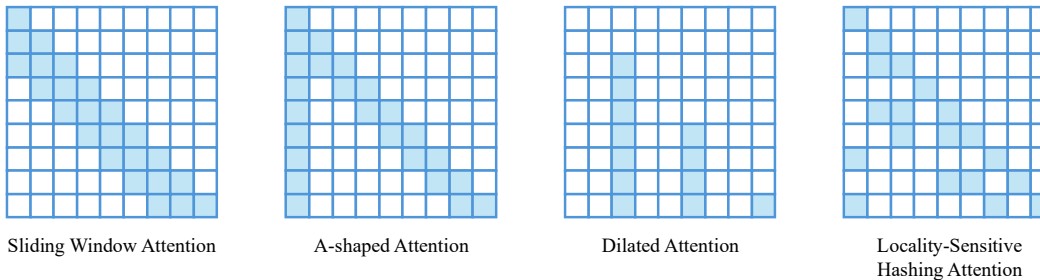

Sliding Window Attention     A-shaped Attention     Dilated Attention     Locality-Sensitive Hashing Attention

Figure 8: Examples for input-dependent and input-independent sparse attention patterns.

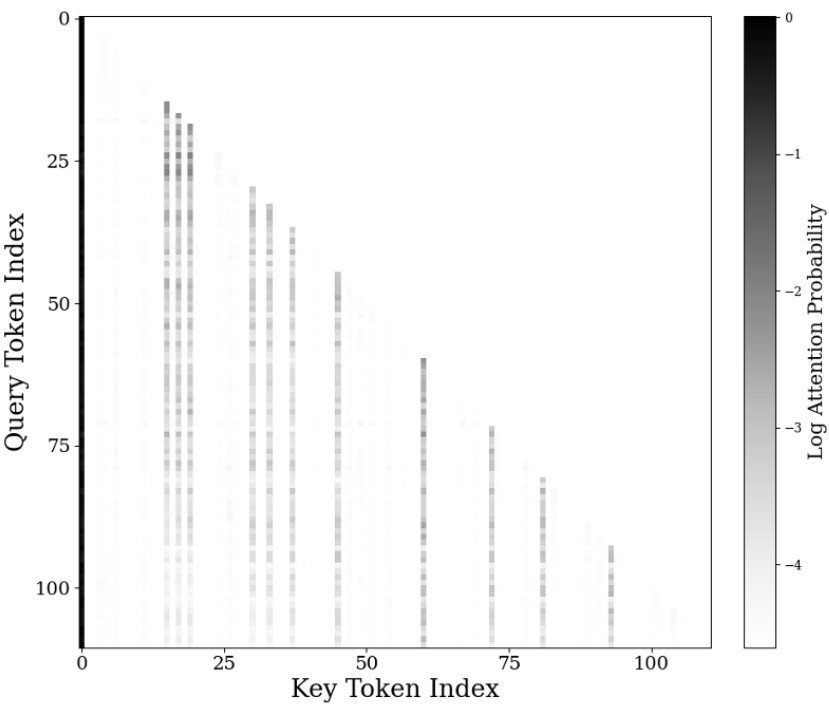

Figure 9: An example of the "vertical-stripe" attention pattern in LLM. We input the first paragraph of the Wikipedia term "Harry Potter" [41] into the Llama 3.2 1B model [13] and visualize the log attention probabilities of the last head in the first layer. The input text is: "Harry Potter is a series of seven fantasy novels written by British author J. K. Rowling. The novels chronicle the lives of a young wizard, Harry Potter, and his friends, Ron Weasley and Hermione Granger, all of whom are students at Hogwarts School of Witchcraft and Wizardry. The main story arc concerns Harry's conflict with Lord Voldemort, a dark wizard who intends to become immortal, overthrow the wizard governing body known as the Ministry of Magic, and subjugate all wizards and Muggles (non-magical people)."

