# OpenReview forum: "Overcoming Long Context Limitations of State Space Models via Context Dependent Sparse Attention"
_NeurIPS.cc/2025/Conference — NeurIPS 2025 poster_

### Official Review · Reviewer_Q4tT · 2025-06-26

**Clarity:** 2
**Significance:** 2
**Originality:** 2
**Rating:** 4
**Confidence:** 5

**Summary:**

The paper presents a novel synthetic task to measure recall capabilities of sequence models at varying complexity levels. The paper then studies how such a simple synthetic recall task can be hardly solved by models based on State Space Models (SSMs) while it can be easily solved by Hybrid models that combine SSMs with the Attention mechanism. The paper also presents an alternative Sparse Attention mechanism inspired by the proposed synthetic recall task and is shown to outperform other variants on small/mid scale experiments.

**Questions:**

1. What is the intuition for using the proposed LSH in section 4.1? The two proposed methods seem arbitrary, why is this the right way to implement the idea of having local hashing?
2. Why using the random matrix H? Do we need to use it at every forward pass or is it initialized and fixed when the model is initialized?
3. Table 1 contains many ablations of the proposed method and it misses comparison with other SoTA methods (like SE-Attention and NSA in [2]). The only comparison is with Samba, however its results are pretty low, why is it the case?
4. Table 2 and 3 is missing baseline numbers from the literature too.

Minor: Line164, broken reference

**Ethical Concerns:**

["NO or VERY MINOR ethics concerns only"]

**Final Justification:**

The authors have been very receptive and engaged during the rebuttal, once the feedback is incorporated, the work will be much more solid and it'll be more impactful for the community.

**Limitations:**

Yes

**Quality:**

2

**Strengths And Weaknesses:**

**Strength**:
1. The paper tackles an important and timely problem: design of optimal subquadratic sequence layers.
2. The introduced synthetic task is simple and could provide complementary information to other commonly used synthetic ones


**Weaknesses**:
1. The synthetic task introduced in this paper should be motivated more and better discussed in light of other commonly used benchmarks. Why do other associative recall tasks fail short? Which properties of trained models cannot be observed on other synthetic recall tasks that can only be observed in this new one? In its current form, it seems to be rather arbitrary, poorly motivated and it misses discussions with previously proposed tasks. For example, why isn’t it possible to use multiple keys with different values per key? This would still test the capability of the model to store all of the information in its state.
2. The local hashing groups keys into bins and lets future queries attend to all keys in the same bin. However this would force the Attention to only look at similar “clusters” with little extra information, this has been shown to reduce expressivity as groups can be aggregated and merged together at reduced information loss. Why is that stage necessary? It’s been shown in SE-Attention (reference [22] in the manuscript) and [2] that it is possible to compress keys while still performing well in downstream tasks, what is the main intuition for not compressing information that is redundant? This is an important modelling assumption which deserves to be precisely motivated and validated.
3. The manuscript misses some key references and comparisons, for example on Hybrid models the paper does not discuss the results in [1] in which authors propose a single unified Hybrid layer capable of compressing redundant information from the past and performing effective recall of old information, their main motivation is similar to the authors’ main ideas and the resulting model is similar to Figure 3, as prior art it should be included in the discussion and compared against. Furthermore, recent work like [2] proposes a similar hardware aware implementation to SE-Attention proposed in [22] (ref in the manuscript), the connection with both these works has not been thoroughly discussed in the manuscript and should be added in the method and the related work section. For example, can the authors comment more on the relationship of SE-Attention and their proposed method?
4. How is the input dependent sparsity pattern implemented? An efficient and scalable implementation requires hardware awareness, but modern hardware cannot efficiently process and move to and from SRAM to HBM memory arbitrary access patterns. See SE-Attn and NSA for some hardware aware implementations, can the authors compare with those in their manuscript?

[1] Zancato L., et al. “B’MOJO: Hybrid State Space Realizations of Foundation Models with Eidetic and Fading Memory”

[2] Yuan J., et al. “Native Sparse Attention: Hardware-Aligned and Natively Trainable Sparse Attention”

---

> ### Author Rebuttal · Authors · 2025-07-31
>
> We sincerely thank the reviewer for the thoughtful and positive assessment of our work. We greatly appreciate your recognition of the significance of our contributions, including the relevance of our study to subquadratic sequence modeling, the value of the joint recall task as a synthetic benchmark, and the motivation behind our proposed hybrid sparse attention mechanism. Your feedback encourages our continued exploration in this important direction. Below, we address your specific questions and concerns in detail.
>
> ---
>
> **W1: On the motivation for proposing joint recall**
>
> Thank you for raising this important point about the motivation of our proposed joint recall task.
>
> The core motivation for joint recall lies in testing *context-dependent* memory, a key aspect of real-world language understanding that is largely absent in prior synthetic tasks such as associative recall. While associative recall benchmarks evaluate whether a model can retrieve a value when queried with a corresponding key, they assume a context-independent one-to-one key-value mapping. This design does not reflect the nature of natural language, where context disambiguates meaning.
>
> To illustrate this, we provide the following example:
>
> - Associative Recall:
>   - Context: “The oceans are blue, and snow is white.”
>   - Query: “What color is snow?”
>   - Answer: “White.”
>
> - Joint Recall:
>   - Context:
>     - “In the U.S., people pay with dollars and drive on the right side of the road.”
>     - “In the U.K., people pay with pounds and drive on the left side of the road.”
>   - Query: “Which side of the road do people drive on in the U.S.?”
>   - Answer: “Right.”
>
>
> From this example, we can see that associative recall does not account for context. Joint recall extends associative recall by incorporating *context-dependency* into key-value associations. For example, while associative recall may map "pay with" to either "dollar" or "pound", joint recall allows it to map to "dollar" in the U.S. and "pound" in the U.K., depending on context. This makes joint recall a more realistic and rigorous synthetic task for both theoretical analysis and empirical benchmark for long-context modeling.
>
> Regarding the suggestion of using multiple keys with different values per key, we agree that such configurations have been explored in benchmarks like Ruler [14]. For reference, we have already tested the Ruler benchmark in our main experiment, including those tasks as well. However, those designs do not evaluate context-sensitive resolution, where the same query token (e.g., “pay with”) may map to different values depending on contextual modifiers (e.g., country). This is precisely the ability we aim to isolate and test with joint recall.
>
> Thus, our task uniquely highlights the limitation of SSMs in capturing input-conditioned compositional associations, and it provides a controlled yet challenging environment for studying improvements from hybrid architectures like LSH+KC.
>
> ---
>
> **W2: On the motivation of introducing LSH**
>
> Thank you for this insightful question. We agree that compressing or merging keys can be beneficial as indicated in related works (e.g., SE-Attention, Native Sparse Attention). Below, we clarify our motivation for introducing LSH attention and discuss its relationship to key compression.
>
> 1. *Theoretical necessity of input-dependent sparsity for hybrid sparse attention.*
>
> We introduce the following *new proposition 4*:
> *There does not exist a 2-layer auto-regressive hybrid model consisting of an SSM layer followed by an input-independent sparse attention layer, which can solve the joint recall task with $o(L^2)$ time complexity, since it requires at least $O(L/k)$ SSM state dimensions, $k$ is the maximum number of keys that each query allowed to attend to in the sparse attention module, as defined in Eq.4.*
>
> The *proof* is provided at the end of the response to this point. This establishes that any sub-quadratic hybrid sparse attention must employ *input-dependent sparse attention*, i.e. sparse attention with a pattern that is conditioned on the query and key representations at inference time.
>
>
> 2. *LSH as an instance of input-dependent sparse attention which solves joint recall.*
>
> We do not claim LSH attention to be the unique or universally optimal input-dependent mechanism. Rather, we show in Proposition 3 that LSH as a concrete instantiation of input-dependent sparse attention, provably suffices to solve joint recall when integrated with SSMs. By choosing LSH, we verify our theoretical result with a concrete, well-studied algorithm. Any other input-dependent scheme with similar properties would also satisfy the theory, and exploring those is an exciting avenue for future work.
>
>
> 3. *LSH’s orthogonality to key compression.*
>
> It is true that compressing keys (via aggregation or learned pooling) can preserve downstream performance in many real-world tasks. In our synthetic joint recall construction, however, no redundancy is present by our construction (please refer to our proof of Proposition 3 in Appendix C.2). For clarity and analytical tractability, we therefore omit key compression from both our proofs and our empirical study. That said, we do not rule out combining LSH+KC with key-compression techniques on real-world data; such hybrid approaches might yield additional efficiency gains.
>
> We hope this clarifies our choice of standard LSH attention: it serves as a concrete and well-established example of input-dependent sparse attention, and when integrated with SSMs, it provides a provable solution to the joint recall task.
>
> *proof of proposition 4.* Consider a key given in the inquiry component of the auto-regressive joint recall task. The model is required to output the associated value token when this key token is provided as input. Taking the query representation from this key token, the sparse attention can attend to at most $k$ key representations from previous tokens, where the key representations are calculated based on the SSM state representations of the first layer. To solve the joint recall task, these $k$ key representations being attended must collectively encode the full context. Since the full context length is $O(L)$, by Corollary 1, $k$ state representations of the generalized SSM in the first layer must use at least $O(L/k)$ dimensions to collectively store the full context. Thus, the per-key computational cost required by the second-layer sparse attention is $O(k \cdot (L/k)) = O(L)$, and therefore the total time complexity is $O(L^2)$.
>
> ---
>
> **W3 / Q3 / Q4: On the related work: B’MOJO, SE-Attn, and NSA**
>
> We would like to clarify that our motivation differs in key ways from the works mentioned. NSA targets hardware efficiency via compression-based sparse attention. SE-Attn improves retrieval in hybrid SSMs through relevance-based memory access. B’MOJO proposes a realization-theoretic architecture for transductive inference. In contrast, our method is motivated by an *expressiveness* analysis grounded in the joint recall task, a synthetic benchmark designed to capture key challenges in *context-dependency* modeling. This analysis reveals a fundamental limitation of SSMs under sub-quadratic constraints. To address it, we introduce a method which combines SSM with input-dependent sparse attention, instantiated by LSH attention, and we present both the theoretical analysis and empirical verification on the advantage of this method.
>
> We also note that none of NSA, SE-Attention, or B’MOJO have released their code. As a result, a direct empirical comparison is currently not feasible. However, we will include a more detailed discussion of these works in the related work section.
>
> ---
>
> **W4: On the hardware-aware implementation**
>
> While we agree that hardware-aware implementation is crucial for scalability and plan to explore it in future work, our current focus is on establishing the theoretical necessity and empirical effectiveness of input-dependent sparse attention. We believe this contribution is significant in its own right, as it identifies a fundamental limitation of SSMs and provides a provably expressive and empirically validated solution.
>
> ---
>
> **Q1: On the hashing implementation**
>
> Our choice to implement both the argmax and sign-bit binning rules is by no means arbitrary, but follows directly from the two most widely adopted LSH-attention schemes in the literature: argmax binning (represented by Reformer [16], Learning-to-hash attention [C.1]), and sign-bit binning (represented by MemoryFormer [C.2], HashAttention [C.3]).
>
> ---
>
> **Q2: On the random hash projection matrix**
>
> Following Reformer [16], we use a random matrix for hash projection. In our implementation, we randomly sample a matrix at each step during training, and fix a random matrix during inference. We do not perform any tuning or adaptation of the random matrix at inference time.
>
> ---
>
> **Q3: On Samba performance**
>
> Thank you for the question. While we have not fully understood the root cause of this discrepancy, we ensured that the comparison was fair by using consistent training settings under the FLA implementation [C.4]. The details of our setup and efforts to ensure fairness are provided in Appendix D.1. We will continue investigating this behavior to better understand the performance gap in future work.
>
> ---
>
> **Minor: broken reference**
>
> We’ll correct it in our future versions.
>
> ---
>
> We once again thank you for your thoughtful and constructive feedback. Your comments have truly helped us clarify our motivation, contributions, and distinctions from prior work, and we appreciate the opportunity to improve our future research as well as the manuscript through your insights.
>
> ---
>
> [C.1] Sparse Attention with Learning to Hash. ICLR 2022.
>
> [C.2] MemoryFormer: Minimize Transformer Computation by Removing Fully-Connected Layers. NeurIPS 2024.
>
> [C.3] HashAttention: Semantic Sparsity for Faster Inference. ICML 2025.
>
> [C.4] fla-org/flash-linear-attention. github.

---

> > ### Author Response · Authors · 2025-08-05
> >
> > Dear Reviewer Q4tT,
> >
> > We appreciate the time and effort you put into reviewing our submission. We would be grateful if you could take a moment to consider our rebuttals and share any additional thoughts you might have. Your insights would be extremely valuable to us.
> >
> > Best,
> > Anonymous authors

---

> > ### Comment · Reviewer_Q4tT · 2025-08-07
> > **Response to the authors**
> >
> > Thank you for the thoughtful rebuttal. The additional exposition on the joint-recall benchmark provides valuable clarifications and it’ll be a valuable addition to the final version of the paper.
> >
> > Overall, I think that even after the rebuttal (and taking into consideration other reviewers’ comments) there are still a few concerns to be addressed.
> >
> > a. [New benchmark validation] There is little evidence that the joint-recall benchmark highlights failure modes not already exposed by other long context benchmarks. This is mostly due to the fact that the current version of the manuscript proposes both the new benchmark and a new model so the space to deeply address both novelties is very limited. In particular, without a multi-model, multi-scale (small and larger pre-trained models, publicly available) it is very hard to judge whether the new benchmark is a good diagnostic or just another new synthetic variant of other tasks in RULER.
> >
> > b. [Comparison with prior work] While it is ok not to compare empirical results at scale if other works have not released code/models. It would be good to try to bridge the gap between the novel model and the previous art at least conceptually, at the moment the comparison remains limited.
> >
> > c. [Effective gains on new results] Reported gains on the 790M model are within 5% relative improvement on a benchmark that’s not saturated. I wonder whether the further complexity of the proposed method compared to the base Mamba baseline justifies its adoption.
> >
> > d. [Answer to YZT8] The present manuscript has been described as “motivated by the goal of improving long-context modeling through a deeper understanding of its expressiveness constraints.” I am wondering whether the empirical evidence showcased is enough (small model sizes and few number of studied architectures).
> >
> > That being said, I really appreciate the authors’ answers and will be willing to support this work further in the review process.

---

> ### Author Response · Authors · 2025-08-08
> **Response to the reviewer Q4tT (part 1)**
>
> We sincerely thank you for the thoughtful and constructive comment. We'll definitely revise our paper for better clarification of joint recall. Below, we address your new concerns in detail.
>
> **a. Empirical validation for the effectiveness of joint recall**
>
> We appreciate the reviewer’s thoughtful concern regarding the diagnostic value of the joint recall benchmark in comparison to existing long-context evaluations such as RULER. Our key motivation for introducing joint recall is to expose a fundamentally different failure mode in long-context models, specifically, **the inability to integrate multiple, spatially distributed contextual cues to recover a value**, rather than simply locating a single relevant key token as in associative recall or RULER.
>
> From the RULER paper, we see that RULER benchmark is already saturated, with multiple models achieving an average accuracy > 90%. However, our *natural language* version of joint recall remains challenging for the most powerful LLM today. Owing to the time constraint of the author-reviewer discussion period, we can only report a case study based on ChatGPT-o3.
>
> In this case study, we consider 5-keys joint recall. The 5 key sets are:
> 1. Years (2020 - 2024);
> 2. Months (January - August);
> 3. Days of the week (Monday - Sunday);
> 4. Time of the day (morning, afternoon and evening);
> 5. Student names (8 common names).
>
> And the value set is 10 common high school subjects. We consider 2 settings to organize the context. The first setting is:
>
> ```
> <input>
> Keep in memory:
> In 2020:
>   In January:
>     On Mondays:
>       In the morning:
>         Susan studied Geography;
>         John studied History;
>         ...
>       In the afternoon:
>         ...
>       In the evening:
>         ...
>     On Tuesdays:
>       ...
>   In February:
>     ...
> In 2021:
>   ...
>
> Now let's recall:
> In June 2023, On Wednesday mornings, Thomas studied:
>
> <answer>
> Mathematics.
> ```
>
> The second setting is:
>
> ```
> <input>
> Keep in memory:
> In January 2020:
>   On Mondays:
>     In the morning:
>       Susan studied Geography;
>       John studied History;
>       ...
>     In the afternoon:
>       ...
>     In the evening:
>       ...
>   On Tuesdays:
>     ...
> In February 2020:
>   ...
>
> Now let's recall:
> In June 2023, On Wednesday mornings, Thomas studied:
>
> <answer>
> Mathematics.
> ```
>
> The only difference between the two settings is that the second collapses the year and month levels (e.g., “January 2020”), while the first retains full hierarchical structure. The key-value associations are randomly sampled once and kept identical across the two settings to ensure comparability. We generate 10 data points for both settings and evaluate ChatGPT-o3’s one-shot response manually via the ChatGPT temporary chat web interface. The data length is about 32K tokens according to the OpenAI online tokenizer.
>
> Surprisingly, we observe that ChatGPT-o3 correctly answers only 5 out of 10 examples in the first setting, but successfully answers all 10 in the second setting. Interestingly, the input length of the second setting is even slightly longer than the first setting, since the year is repeated after each month.
>
> From this case study, we draw two preliminary conclusions: (1) even today’s most powerful LLMs are not yet capable of perfectly solving the *natural language* version of joint recall task; and (2) their performance is limited not just by context length, but also by their ability to integrate and compose multiple pieces of context effectively.
>
> Therefore, joint recall is not just another synthetic variant of tasks in RULER. RULER mainly focuses on context length, we are among the first to show that **context structure** also matters. Note that this point is not just about the limitations of SSMs. It can also provide insight for the broader field of long-context modeling.
>
> We sincerely thank the reviewer for encouraging us to further investigate this direction, and we will report more formal and comprehensive experiments in the final version of our paper.

---

> ### Author Response · Authors · 2025-08-08
> **Response to the reviewer Q4tT (part 2)**
>
> **b. Conceptual comparison with compression-based sparse attention**
>
> Typically, compression-based sparse attention rely on two key assumptions:
> 1. The input context is compressible, and
> 2. The context exhibits predefined structural properties (e.g., block boundaries for block compression), which is suitable for compression.
>
> In contrast, we do not assume that the context is compressible or that it follows a known structure. Instead, we focus on improving a model’s general ability to compose multiple, spatially distributed cues in the context. This is theoretically grounded by the joint recall task, where we show that pure SSMs face fundamental limitations, and where our hybrid LSH+KC mechanism overcomes those limitations by introducing input-dependent sparse attention. Due to these differences in assumptions and goals, we do not view compression-based models as directly comparable baselines in our study.
>
> A minor point on technique: we note that many prior works (including NSA, SE-Attn, Samba, and B’MOJO) incorporate sliding window attention as a core component. However, in our study, both theoretical analysis and empirical results suggest that sliding window attention might not be the best solution for hybrid sparse attention.
>
> Last but not least, we agree that conceptual comparison is important. We will revise the manuscript to include a more thorough discussion of these works in the related work section.
>
> ---
>
> **c. Effective gains on new results**
>
> **d. Response to Reviewer YZT8: Scope of empirical support**
>
> We thank the reviewer for raising these important concerns regarding the scale and impact of our empirical results.
>
> First, we acknowledge that the performance gains reported on the 790M model are modest. However, we would like to clarify that the 790M model under continual pre-training is still far from convergence. Due to the formatting constraints in this rebuttal phase, we are unable to include the training loss curves, but we observed a consistent downward trend that suggests further improvements with continued training.
>
> Second, as in the original Mamba paper, we followed the recommended practice of using a smaller learning rate for continual pre-training at this scale. While this promotes stable learning, it also increases the number of training steps required for meaningful gains, posing a significant computational burden under academic resource constraints.
>
> Third, it's important to highlight that LongBench is a challenging benchmark where even well-designed architectural changes often yield limited gains. As we noted in our response to Reviewer YZT8’s comment W1, even a 130M hybrid model augmented with *full* attention achieves only marginal improvements over the Mamba baseline. This underscores the difficulty of improving LongBench performance, especially in continual pre-training settings.
>
> Fourth, and more broadly, we would like to clarify that the primary goal of our empirical study is to validate the our theoretical analysis. Across all our settings, spanning both 130M and 790M models, The Pile and TxT-360 continual pre-training datasets, and evaluations on RULER, LongBench, as well as the joint recall task, we **consistently observe the following results:**
>
> 1. **Adding input-independent sparse attention (e.g., sliding window) to SSMs results in negligible gains.**
> 2. **Adding only LSH or only KC attention is similarly ineffective.**
> 3. **Only the combination of LSH and KC attention yields meaningful improvements, in line with our expressiveness analysis.**
>
> Lastly, we acknowledge the reviewer’s concern about limited architecture diversity. Ideally, we would explore from-scratch pre-training as the most principled route to test generalization, but this is beyond our current academic resources. As a result, we rely on continual pre-training, which is inherently constrained by the properties of the base checkpoint (e.g., Mamba’s 2K pre-training context length). Despite this, we believe our findings offer general valuable insight into the design of sub-quadratic hybrid architectures and help guide future exploration at scale.
>
> ---
>
> Once again, we sincerely thank you for your thoughtful and detailed feedback. Your comments have helped us clarify our contributions, better position our work within the literature, and identify valuable directions for further improvement. We will incorporate the suggested revisions in the final version of the manuscript.

---

> > ### Comment · Reviewer_Q4tT · 2025-08-09
> > **Acknowledgment**
> >
> > I thank the authors for their further clarifications, I think that revising the manuscript taking into considerations the suggestions in this rebuttal will make this work much more solid and impactful to the community. I'll raise my score accordingly. Thanks!

---

> > > ### Author Response · Authors · 2025-08-09
> > > **Thank you for your suggestions!!**
> > >
> > > We sincerely thank the reviewer for the thoughtful and constructive feedback, which has helped us strengthen both the contextual positioning and the methodological framing of our work. Your insightful suggestions, such as empirically motivating the joint-recall benchmark and discussing connections with prior work like SE-Att, NSA and B'MOJO, have been especially valuable. We appreciate your engagement and willingness to support the paper, and will incorporate these insights to further improve the final manuscript.

---

### Official Review · Reviewer_YZT8 · 2025-07-05

**Clarity:** 1
**Significance:** 3
**Originality:** 3
**Rating:** 4
**Confidence:** 4

**Summary:**

This paper addresses the limitations of State-Space Models (SSMs) in capturing complex long-range dependencies that require integrating multiple contextual cues. The authors introduce a new synthetic task, "joint recall," to demonstrate this weakness and propose a hybrid sparse attention mechanism combining Locality-Sensitive Hashing (LSH) and a Key-Centered (KC) attention. Empirical results on both synthetic and real-world NLP benchmarks show that augmenting Mamba with this hybrid attention significantly improves its long-context modeling performance.

**Questions:**

1. The QA2 task results are missing for RULER benchmark. Could you add them?

2. Could you add a baseline that use dense attention instead of sparse attention for hybridization?

3. Which dataset did you use for continual pretraining?

4. How sensitive are the continual training results on the scalers of the auxilary loss in KC attention?

5. How many parameters doe the MLP have for key scoring?

6. Could you also add a setting of SW+LSH, so that we can better understand the effectiveness of KC?

7. How many test samples are in the joint recall synthetic dataset? Could you include the standard deviation for the evaluation on synthetic tasks?

**Ethical Concerns:**

["NO or VERY MINOR ethics concerns only"]

**Final Justification:**

The added results mostly resolved my concerns.

**Limitations:**

yes

**Quality:**

2

**Strengths And Weaknesses:**

Strengths:

1. The proposed LSH+KC attention is thoughtfully designed. It combines the content-based clustering of LSH with a key selection mechanism (KC attention) specifically created to handle "global hub" tokens (like instructions), addressing a known weakness of simple LSH.

2. The idea of using both key representation and historical query representaitons for key importance scoring is novel. And using ranking loss on downsampled number of keys to learn the scoring module drastically reduces the training computation costs, and is well motivated.

Weaknesses:

1. The paper seems rushly written. Experiment results are not complete, and the experiment setup and evaluation settings are missing. Technical details on KC attention are not clear. The results on LongBench seem not significant.

2. The proposed KC attention has similar performance as sliding window attention on RULER. It is unclear how effective KC it is in real-world scenario.

3. The scale of the experiments are small. The model trained on synthetic data are only two layers and the 130M sized Mamba model are continually pretrained for the results on real world data.

4. Missing Citations. The authors should discuss with previous works on dynamic sparse attention, especially those also select QK representations [1,2], to better position their works.


---

[1] Faster Causal Attention Over Large Sequences Through Sparse Flash Attention

[2] Sparse Modular Activation for Efficient Sequence Modeling

---

> ### Author Rebuttal · Authors · 2025-07-31
>
> We sincerely thank you for the thoughtful and encouraging feedback. We are glad that you found the design of our LSH+KC attention mechanism to be novel and well-motivated.
>
> Below, we address your specific questions and concerns in detail.
>
> ---
>
> **W3: The scale of the experiments are small.**
>
> We have newly conducted continual pre-training based on the 790M Mamba official checkpoint. These larger-scale experiments confirm the effectiveness of our proposed LSH+KC method on real-world long-context benchmarks. For these experiments, we use the TxT-360 corpus (not the same as The Pile used in our paper) as the continual pre-training dataset. We continual pre-train for 50,000 steps with a context length of 2K, which is followed with instruction tuning using UltraChat (the same as in the paper) for 10,000 steps with a context length of 2K. We also train new 130M models in this setting. The new results on Ruler and LongBench are presented below:
>
> **Table R.1: 4K Ruler results based on 790M Mamba**
>
> |Model|NIAHS1|NIAHS2|NIAHS3|NIAHMK1|NIAHMK2|NIAHMK3|NIAHMV|NIAHMQ|VT|CWE|FEW|QA1|QA2|Average|
> |-|-|-|-|-|-|-|-|-|-|-|-|-|-|-|
> |Mamba|100|19.2|9.2|5.2|0.6|0|0.65|1.85|5.72|4.3|100|26.2|78.6|27.0|
> |+D|0|0|0|0|0|0|0|0|65.2|0|100|0|73|18.3|
> |+SW|100|18.2|5|5.6|0.2|0.2|1.75|2.1|4.4|4.78|100|22.4|78.8|26.4|
> |+SW+D|100|20.2|8.2|6.2|0.4|0.2|1.45|1.25|4.36|4.32|100|24.6|79|26.9|
> |+SW+SK|100|19.8|7.2|6.6|1.2|0.2|1.3|0.8|4.36|3.24|100|24.2|79|26.8|
> |+LSH|0|0|0|0|0|0|0|0|0|0.04|100|0.2|73|13.3|
> |+KC|99.8|20.2|7.8|8.2|1|0.2|1.2|1.85|9.32|2.86|100|22.6|78.6|27.2|
> |+LSH+KC|100|23|11|9|0.8|0|3.4|2.75|6.36|2.54|100|29|78.8|**28.2**|
>
> **Table R.2: LongBench results based on 790M Mamba**
>
> |Model|2WikiMQA|GovReport|HotPotQA|LCC|MultiNews|MultiFQA|MuSiQue|NQA|PsgCnt|PsgRet|Qasper|QMSum|Repobench|SamSum|Trec|TriviaQA|Average|
> |-|-|-|-|-|-|-|-|-|-|-|-|-|-|-|-|-|-|
> |Mamba|13.39|20.3|8.37|39.51|20.36|27.62|3.89|8.35|1.04|1.5|10.69|20.36|38.47|20.15|45|37.59|19.79|
> |+D|0|0.21|0|2.87|0.26|0|0|0|0|0|0|0.72|3.46|0.7|0|0|0.51|
> |+SW|13.81|20.58|9.25|38.06|20.64|27.55|5.15|8.12|1.2|1.5|10.55|19.57|38.38|22.42|43|36.55|19.77|
> |+SW+D|13.6|20.39|8.55|39|20.74|27.55|4.33|7.98|0.83|1.56|10.17|19.73|38.84|22.49|43|37.02|19.74|
> |+SW+SK|13.63|20.77|8.55|38.59|20.2|27.31|4.56|7.66|0.27|1.74|11.32|20.09|38.5|20.73|44.5|37.16|19.72|
> |+LSH|1.66|1.49|1.5|7.74|0.98|2.32|0.96|0.87|0.1|0|2.3|5.05|7.13|3.89|0|0.9|2.31|
> |+KC|12.93|20.93|9.34|38.22|20.83|26.56|4.34|7.46|0.27|1.67|11.65|19.35|38.17|22.97|45|37.52|19.83|
> |+LSH+KC|12.73|20.64|8.79|39.3|21.79|28.31|3.95|7.64|0.83|2|11.94|19.63|39.07|21.04|43|38.47|**19.95**|
>
> **Table R.3: LongBench results based on new 130M Mamba**
>
> |Model|Average|
> |-|-|
> |Mamba|11.40|
> |+D|0.82|
> |+SW|11.52|
> |+SW+D|10.98|
> |+SW+SK|11.28|
> |+LSH|2.73|
> |+KC|10.99|
> |+LSH+KC|**11.58**|
>
> ---
>
> **W1: The paper seems rushly written.**
>
> We sincerely apologize for that. After submission, we have substantially revised the manuscript to improve clarity, organization, and technical precision.
>
> **W1: Experiment results are not complete; Q1: Add the QA2 task in Ruler benchmark.**
>
> Thank you for pointing this out. The results are provided at Table R.1. These 790M models were pre-trained with a 2K context length, so the 4K evaluation tests their extrapolation ability. Notably, our LSH+KC architecture still outperforms all baselines on average Ruler performance (including QA2), highlighting its robustness to longer contexts.
>
> **W1: Experiment results are not complete; Q2: Add the hybrid dense attention baseline. Q6: Add the SW+LSH baseline. Q8: Include the standard deviation on joint recall.**
>
> Across all benchmarks, our proposed Mamba+LSH+KC consistently outperforms Mamba+LSH+SW and narrows the gap between SSM and hybrid full attention. For the joint recall experiments, due to the rebuttal character limitations, please refer to our rebuttal of W2/Q2 to the reviewer MdxV. The Ruler results are provided in the following table:
>
> |Model|NIAHS1|NIAHS2|NIAHS3|NIAHMK1|NIAHMK2|NIAHMK3|NIAHMV|NIAHMQ|VT|CWE|FEW|QA1|Average|
> |-|-|-|-|-|-|-|-|-|-|-|-|-|-|
> |Mamba|99.6|53.4|7.4|20.8|0|0|17.8|4.8|1.5|1.3|100|14|26.7|
> |Mamba+LSH+SW|99.8|79.2|23.6|23|0|0|18.4|8.35|2.96|1.3|100|11.8|30.7|
> |Mamba+LSH+KC|100|92.4|34.6|24|0.2|0|20.4|3.8|4.9|1.7|100|12.8|32.9|
> |Mamba+FullAttn|100|100|57.4|35.8|1.2|5.4|31.35|10.55|2.04|4.88|100|15.6|38.69|
>
> And the LongBench results are provided in the following table:
>
> |Model|2WikiMQA|GovReport|HotPotQA|LCC|MultiNews|MultiFQA|MuSiQue|NQA|PsgCnt|PsgRet|Qasper|QMSum|Repobench|SamSum|Trec|TriviaQA|Average|
> |-|-|-|-|-|-|-|-|-|-|-|-|-|-|-|-|-|-|
> |Mamba|6.11|15.16|3.35|34.57|16.73|12.72|2.51|3.02|0.85|0.5|4.97|16.49|35.76|1.67|10.5|12.76|11.10|
> |Mamba+LSH+SW|6|14.17|2.95|34.65|15.58|13.65|2.13|2.28|0.41|0.75|5.01|13.71|36.24|1.91|14.5|12.82|11.04|
> |Mamba+LSH+KC|6.71|15.47|3.34|34.81|14.95|13.64|1.76|2.6|1.64|0.3|5.54|14.22|35.47|1.89|14|15.17|11.34|
> |Mamba+FullAttn|5.46|16.99|3.16|34.26|17.85|13.28|2.2|2.65|1.14|0|5.74|16.23|35.96|1.73|14.5|14.27|11.58|
>
> **W1: The experiment setup and evaluation settings are missing.**
>
> We apologize if the presentation was unclear and would like to clarify that:
> - We describe the baseline setup and evaluation procedure in Section 5, with additional training and implementation details provided in Appendix D.
> - For the joint recall task, evaluation settings are detailed in Section 5.1.
> - For Ruler [14] and LongBench [3], we follow their standard evaluation settings.
>
> **W1: Technical details on KC attention are not clear.**
>
> We provide the technical description of KC attention in Section 4.2, including the method, mathematical formulation, and training objective. The MLP is the simplest 3-layer MLP with ReLU activation.
>
> **W1: The results on LongBench seem not significant.**
>
> We emphasize that LongBench poses significant challenges for long-context modeling and demands substantial training to perform well. Our largest continual pretraining run, a 790M model trained on approximately 8B tokens, remains far from convergence. Nevertheless, our proposed LSH+KC architecture consistently outperforms all baselines on LongBench across three evaluation settings (Table 3, R.2, R.3), underscoring its practical effectiveness even under limited academic training resources.
>
> ---
>
> **W2: It is unclear how effective KC it is in real-world scenario.**
>
> We would like to clarify that KC attention is not intended to serve as a stand-alone module, but rather as a complementary component to enhance the expressiveness of LSH attention. As shown in our experiments (Tables 2, 3, R.1, R.2, R.3), our hybrid LSH+KC attention consistently outperforms both the base SSMs and all other SSM variants augmented with input-independent sparse attention, including sliding window attention, A-shaped attention (sliding window + sink) and sliding window + dilated attention. This advantage holds across both synthetic and real-world long-context benchmarks, validating the effectiveness of KC as part of a larger input-dependent attention design.
>
> ---
>
> **W4: Missing Citations.**
>
> The two referenced papers contribute to sparse attention from an efficiency perspective. The first presents a Triton-based kernel that accelerates both QK-sparse and hash-sparse attention. The second introduces Sparse Modular Activation, which dynamically gates attention modules per token in hybrid SSM-attention models.
>
> In contrast, our work is motivated by the goal of improving long-context modeling through a deeper understanding of its *expressiveness* constraints. First, building on a theoretical analysis of the joint-recall task, we identify a fundamental limitation in the *representational capacity* of SSMs and motivate the integration of LSH attention to address this challenge. Second, we further propose KC attention to address the *expressiveness* limitation of LSH. Finally, we integrate our method into modern state-space models, specifically Mamba and Mamba-2, demonstrating its empirical benefits at scale.
>
> We appreciate the reviewer’s pointers to these works and will add a subsection in Related Work to discuss related sparse attentions.
>
> ---
>
> **Q3: Which dataset did you use for continual pretraining?**
>
> We followed the original Mamba setup and used The Pile for continual pretraining, as noted in Appendix D.2.
>
> ---
>
> **Q4: How sensitive are the continual training results on the scalers of the auxiliary loss in KC attention?**
>
> Thank you for bringing this up. While we did not extensively tune this scaling factor, we observed that choosing a suitable value is important for achieving good downstream performance. Here are the LongBench results based on different scaler values:
>
> |$\alpha$|2WikiMQA|GovReport|HotPotQA|LCC|MultiNews|MultiFQA|MuSiQue|NQA|PsgCnt|PsgRet|Qasper|QMSum|Repobench|SamSum|Trec|TriviaQA|Average|
> |-|-|-|-|-|-|-|-|-|-|-|-|-|-|-|-|-|-|
> |0.05|5.3|14.62|2.63|35.62|13.4|11.42|1.82|1.92|0.66|1.08|4.73|12.5|36.73|1.63|15.5|14.11|10.85|
> |0.1|6.71|15.47|3.34|34.81|14.95|13.64|1.76|2.6|1.64|0.3|5.54|14.22|35.47|1.89|14|15.17|11.34|
> |0.2|5.56|15.23|3.18|35.78|14.01|11.98|1.78|2.03|0.66|1.17|4.83|12.89|36.29|1.66|15|14.6|11.04|
>
> ---
>
> **Q5: How many parameters does the MLP have for key scoring?**
>
> The MLP used for key scoring in KC attention is lightweight relative to the base model. We provide the comparison of the number of parameters in the following table:
>
> |Officially Released Mamba|MLP for KC attention|Mamba+LSH+KC|
> |-|-|-|
> |129.1M (100%)|0.5M (0.39%)|130.9M (101.33%)|
> |793.2M (100%)|1.0M (0.13%)| 799.1M (100.74%)|
>
> This demonstrates that LSH+KC attention introduces minimal overhead while delivering meaningful performance improvements.
>
> ---
>
> **Q7: How many test samples are in the joint recall synthetic dataset?**
>
> As noted in Line 232, the joint recall synthetic dataset contains 10,000 test samples.
>
> ---
>
> We sincerely thank the reviewer once again for the constructive feedback and insightful questions, which have helped us strengthen the clarity, depth, and presentation of our work.

---

> ### Author Response · Authors · 2025-08-08
> **Thank you for your response!!**
>
> We sincerely thank the reviewer for the thoughtful and detailed feedback, as well as for increasing the score. Your suggestions, particularly on running more comprehensive experiments and improving writing clarity, are deeply appreciated. We are committed to revising the paper based on your advice and are grateful for your engagement and support.

---

### Official Review · Reviewer_MdxV · 2025-07-06

**Clarity:** 3
**Significance:** 3
**Originality:** 3
**Rating:** 4
**Confidence:** 5

**Summary:**

This paper addresses the limitations of state-space models (SSMs) in long-context natural language processing (NLP) by introducing a novel hybrid sparse attention mechanism. The authors propose a new synthetic task, joint recall, to evaluate the model's ability to integrate multiple contextual cues. Through theoretical analysis and empirical evaluation, they demonstrate that their proposed hybrid sparse attention mechanism, combining locality-sensitive hashing (LSH) and key-centered (KC) attention, significantly improves the long-context modeling capacity of SSMs. The paper also validates these findings on real-world NLP tasks, showing enhanced performance on benchmarks such as Ruler and LongBench.

**Questions:**

Apart from the weaknesses mentioned above, there might be some questions that would benefit from this manuscript.

* (Q1) **Broader Impacts.** Could the authors provide a discussion on the potential societal impacts of their work, including any negative consequences and mitigation strategies?

* (Q2) **Statistical Significance.** How do the authors plan to address the lack of error bars or other measures of statistical significance in their experimental results?

**Ethical Concerns:**

["NO or VERY MINOR ethics concerns only"]

**Final Justification:**

After considering the rebuttal from the authors and review comments from other reviews, the authors have tackled my concerns, and I decided to keep my original score.

**Limitations:**

The paper is technically sound, with a clear and innovative approach to addressing long-context limitations in SSMs. The theoretical and empirical evaluations are comprehensive, supporting the claims made. The authors have acknowledged some limitations in the appendix, but a more explicit discussion in the main paper would be beneficial. Additionally, the paper would benefit from a clearer discussion on the potential societal impacts and future directions of their work.

**Quality:**

3

**Strengths And Weaknesses:**

### Strengths

* **(S1) Innovative Approach**: The paper introduces a novel hybrid sparse attention mechanism that addresses the limitations of SSMs in
capturing long-range dependencies, demonstrating both theoretical and empirical improvements.

* **(S2) Synthetic Task**: The introduction of the joint recall task provides a unique framework for evaluating the model's ability to integrate multiple contextual cues, enhancing the interpretability of the results.

* **(S3) Empirical Validation**: The paper provides comprehensive empirical evidence supporting the effectiveness of the proposed method on both synthetic and real-world datasets, showcasing its practical utility.

* **(S4) Technical Rigor**: The theoretical analysis is thorough, with clear assumptions and proofs, ensuring the soundness of the proposed approach.

### Weaknesses

* **(W1) Limited Discussion on Broader Impacts**: The paper does not discuss potential societal impacts, which is a missed opportunity to address broader ethical considerations.

* **(W2) Statistical Significance**: Firstly, the paper lacks reporting of error bars or other measures of statistical significance, which could strengthen the credibility of the experimental results. Secondly, expand to explicitly contrast joint recall with positional-stress protocols (e.g., ‘lost-in-the-middle’) and multi-key associative tasks.

* **(W3) Reproducibility**: While the paper mentions the availability of code and data, it does not provide explicit instructions or URLs, potentially hindering full reproducibility.

* **(W4) Missing hardware-aligned sparse baselines:** More comparisons to relevant previous works and reporting of throughput/latency and memory utilization to substantiate “sub-quadratic” *in practice* would enhance the manuscript with more convincing evidence of the proposed method.

---

> ### Author Rebuttal · Authors · 2025-07-31
>
> We sincerely thank you for your thoughtful evaluation of our work. We greatly appreciate the positive feedback regarding the novelty of our hybrid sparse attention, the clarity of our theoretical analysis, and the solidity of the empirical results. We are also glad that the introduction of the joint recall task was seen as a meaningful contribution to theoretical analysis and empirical benchmark for long-context modeling. Below, we address your specific questions and concerns in detail.
>
> ---
>
> **W1/Q1: On the potential societal impacts of our work**
>
> Thank you for raising this important point regarding broader societal impacts.
>
> Our work focuses on fundamental architectural improvements for long-context sequence modeling, particularly by enhancing the theoretical and empirical understanding of hybrid sparse attention in state-space models. The goal is to advance the efficiency and expressiveness of machine learning models at the architectural level, a domain that is general-purpose and infrastructure-oriented rather than application-specific.
>
> As such, our contributions do not involve direct interaction with human users, personal data, or deployment in potentially risky domains (e.g., medical, legal, or surveillance applications). Consequently, we believe our work does not introduce significant ethical risks or societal concerns in its current scope.
>
> Meanwhile, we agree that architectural advances may eventually be used in downstream applications. Therefore, we fully support the community’s efforts to consider responsible deployment and fairness at later stages of the ML pipeline. We have updated the Conclusion section to explicitly clarify the scope and neutrality of our contributions with respect to societal impacts.
>
> ---
>
> **W2/Q2: Statistical significance report**
>
> We appreciate your suggestion regarding the inclusion of statistical significance measures.
>
> For the Ruler [14] and LongBench [3] benchmarks, we follow the established evaluation protocol used in prior work, which typically reports single-run results without error bars. We adopted the same reporting standard to ensure comparability and consistency.
>
> For our proposed joint recall task, we conducted experiments using three random training seeds and summarize the average results with standard deviations in the table below:
>
> ||Mamba|Mamba2|
> |-|-|-|
> |Base|16.3 $\pm\$ 0.7|36.6 $\pm\$ 2.7|
> |+D|7.8 $\pm\$ 0.8|19.6 $\pm\$ 16.3|
> |+SW|18.7 $\pm\$ 4.5|70.6 $\pm\$ 46.8|
> |+SW+D|16.6 $\pm\$ 1.2|48.6 $\pm\$ 46.2|
> |+SW+SK|16.4 $\pm\$ 0.1|49.3 $\pm\$ 42.7|
> |+LSH|11.6 $\pm\$ 2.5|13.5 $\pm\$ 12.6|
> |+LSH+SW|16.5 $\pm\$ 0.2|50.7 $\pm\$ 43.2|
> |+KC|36.6 $\pm\$ 35.5|60.1 $\pm\$ 34.1|
> |+LSH+KC|38.0 $\pm\$ 37.3|74.3 $\pm\$ 35.0|
> |+FullAttn|48.4 $\pm\$ 45.2|100.0 $\pm\$ 0.0|
> |Samba|6.3 $\pm\$ 0.1|/|
>
> Two new baselines are added to this table: hybrid LSH + sliding window (SW) attention and hybrid full attention. As observed, the variance in performance can be relatively high. This behavior is inherent to the nature of the synthetic task: the model either learns the underlying rule and achieves near-perfect accuracy, or fails to do so and performs poorly when being stuck at a local minimum. This leads to a bimodal outcome distribution, which inflates the standard deviation.
>
> ---
>
> **W3: Data availability and code reproducibility**
>
> We have provided the link to the full codebase at multiple locations in the paper (Lines 248, 479, and 518), and we will further highlight these links in the final version for better visibility.
>
> Both the joint recall synthetic task and the continual pre-training experiments are fully reproducible by running the scripts in our repository. For the continual pre-training experiments in the paper, we use only publicly available datasets:
>
> - Continual pre-training follows Mamba [10] using The Pile dataset.
> - Instruction tuning follows Mamba-Chat [20] using UltraChat.
> - Evaluation is performed on Ruler [14] and LongBench [3], both of which are publicly accessible.
>
>
> We provide a detailed description of training setups in Appendix D. We hope this clarifies the reproducibility of our experiments.
>
> ---
>
> Once again, we sincerely thank you for the helpful comments and suggestions. We sincerely hope that our rebuttal resolves the concern you highlighted, and we are grateful for your time and insights.

---

> ### Author Response · Authors · 2025-08-05
>
> Dear Reviewer MdxV,
>
> We appreciate the time and effort you put into reviewing our submission. We would be grateful if you could take a moment to consider our rebuttals and share any additional thoughts you might have. Your insights would be extremely valuable to us.
>
> Best,
> Anonymous authors

---

> ### Comment · Reviewer_MdxV · 2025-08-08
> **Feedback to Authors' Rebuttal**
>
> Thanks for the detailed response and the efforts the authors provided. The rebuttal addressed most concerns satisfactorily. Here are some further suggestions.
>
> * More discussion with related work. The authors could further discuss or add these baselines, like SE-Attn [1], B’MOJO [2], and NSA [3]. Given their direct relevance (hybrid span expansion and hardware-aligned sparsity), additional comparisons would make the manuscript more convincing with claims of designed methods, e.g., the hardware-aligned sparse methods in NSA are similar to the proposed method.
>
> * LSH design choices. Do the authors further explore or clarify whether the random matrix HH is fixed vs. resampled, the number of hash rounds/bins, or bucket balancing—key for stability and reproducibility in LSH attention.
>
> * More Comparison on popular real-world benchmarks. In the manuscript and rebuttal responses, the authors have provided benchmark results on Ruler and LongBench, which are convincing to verify the efficiency. The authors could further provide commonly used benchmarks like Wikitext, common-sense reasoning tasks, and question-answering tasks, as conducted in previous correlated works like [4].
>
> ### Reference
>
> [1] Expansion Span: Combining Fading Memory and Retrieval in Hybrid State Space Models. NeuS, 2025.
>
> [2] B'MOJO: Hybrid State Space Realizations of Foundation Models with Eidetic and Fading Memory. NeurIPS, 2024.
>
> [3] Native Sparse Attention: Hardware-Aligned and Natively Trainable Sparse Attention. ACL, 2025.
>
> [4] Gated Linear Attention Transformers with Hardware-Efficient Training. ICML, 2024.

---

> ### Author Response · Authors · 2025-08-09
> **Thank you for your suggestions.**
>
> We thank the reviewer for the constructive feedback and helpful suggestions:
> - Related Work and Baselines. We thank the reviewer for pointing out SE-Attn [1], B’MOJO [2], and NSA [3]. We agree that these works are relevant and will revise the manuscript to include a more thorough discussion of them in the related work section. We will also provide empirical comparisons once their official implementations become publicly available.
> - LSH Design Choices. We appreciate the suggestion to clarify our LSH configuration. In our implementation, the random projection matrix $H$ is resampled at each training step to improve robustness, and fixed during inference to enable consistent auto-regressive cache representation. We use a single hash round with $k$ hash bins, where $k$ is the sparsity constraint (the maximal number of keys a query can attend to) to maintain efficiency. While identifying the optimal LSH setup is not the primary focus of this work, we will explore these design variants (e.g., more rounds or bucket balancing) when computational resources permit.
> - Additional Benchmarks. If the reviewer is referring to the benchmark in Table 2 of GLA [4], we note that we have already included the same evaluation in Appendix E. We did not place it in the main text because these benchmarks have relatively short context lengths, whereas our focus is on long-context modeling. Nevertheless, we will make this connection explicit in the revision so that readers can easily locate the results.
>
> We again sincerely thank the reviewer for the thoughtful and detailed feedback. We are committed to revising the paper based on your advice and are grateful for your engagement and support.

---

### Author Response · Authors · 2025-08-09
**General Response and Thank You All**

Dear Reviewers, AC, SAC, and PC,

We sincerely thank you for your time, feedback, and engagement throughout the review process. We appreciate the effort you’ve dedicated not only to the initial reviews but also to the follow-up rebuttal and open discussion. We are encouraged that every reviewer has now raised their score to 4 or higher.

Our **main contributions** are summarized as follows:

1. **Joint recall task**: We introduce joint recall, a novel synthetic task that extends associative recall to context-dependent key–value association. This task offers a new perspective for both theoretical analysis and empirical benchmarking of long-context modeling.

2. **Theoretical analysis**: We show that integrating state-space models (SSMs) with *input-dependent* sparse attention has the expressiveness to solve multi-query joint recall with sub-quadratic computation.

3. **Novel architecture**: Guided by this theoretical insight, we propose a new architecture that consistently outperforms both standalone SSMs and SSMs integrated with *input-independent* sparse attention on synthetic and real-world long-context benchmarks.

We are grateful for the **reviewers' recognition** of our contributions, especially:

- **Reviewer Q4tT** – for highlighting the importance of studying sub-quadratic architectures for sequence modeling and recognizing the value of our joint recall task.

- **Reviewer YZT8** – for acknowledging the novelty of our proposed architecture and the corresponding training algorithm.

- **Reviewer MdxV** – for fully recognizing our contributions, including the novel synthetic task, thorough theoretical analysis, and comprehensive empirical evaluation.

We believe we have **addressed the main concerns** regarding:

- **Motivation of joint recall** – which extends associative recall to be conditioned on context, testing an important long-context capability: combining multiple cues in context. This capability has not been effectively addressed even by the most powerful LLM today, ChatGPT-o3.

- **Relation to existing work** – we have compared our work with contemporary approaches and clarified our uniqueness in terms of goal, motivation, theoretical assumptions, and methodology.

- **Model scale and performance** – we now provide results for a 790M model. Across different model scales, training data, and evaluation benchmarks, we consistently observe the effectiveness of our method.

We will revise the camera-ready version to:

- Clarify the motivation of joint recall along with the empirical evidence we presented in our discussion with Reviewer Q4tT.

- Clarify the goal of this study: improving state-space models long-context modeling through a deeper understanding of its expressiveness constraints.

- Add a more detailed discussion of related work on sparse attention.

- Clarify our contributions and discuss potential societal impacts.

We will also extend the following in future work:

- Conduct a sufficient number of experiments to explore best practices for architecture design.

- Further scale up the model.

- Perform hardware-aware optimizations.

Finally, we emphasize that our joint recall task is not designed specifically for SSM optimization, but could serve as a **general-purpose tool for long-context modeling studies**.

Best, Anonymous authors

---

### Note · Authors · 2025-08-13

We have summarized the rebuttal progress in the following General Response. We sincerely thank all reviewers, the AC, SAC, and PC for their time, thoughtful feedback, and constructive engagement throughout the review process. We will carefully revise the final version to incorporate every suggestion provided, ensuring the paper fully reflects the valuable insights from this review process.

---

### Decision · Program_Chairs · 2025-09-17

**Decision:**

Accept (poster)

**Comment:**

The paper argues that to achieve sub-quadratic long-context modeling with State-Space Models (SSMs), it needs input-dependent sparse attention. It introduces "joint recall," a novel synthetic task testing context-dependent key-value associations. And based on this task, It formalizes the assumption (SSMs need needs input-dependent sparse attention) via an expressiveness argument (e.g., a proposition showing input-independent sparsity is insufficient under sub-quadratic constraints).

Guided by this analysis, the authors propose a hybrid mechanism LSH + Key-Centered attention:
LSH attention for content-addressable retrieval with query/key-conditioned sparsity.
Key-Centered attention to emphasize “global hub” tokens (e.g., instructions), with a lightweight scoring MLP trained via a ranking objective.

Empirical evaluation is done on Joint Recall, RULER, and LongBench, including continual pre-training of Mamba/Mamba-2 at 130M and 790M scales. Reported results show consistent gains for LSH+KC over SSM baselines and SSMs with input-independent sparsity.

Strengths
1. The expressiveness argument well motivates the necessary of input-dependent sparsity
2. Joint Recall is a useful task that isolates context-dependent composition
3. Across synthetic and real benchmarks, LSH+KC outperforms pure SSMs and SSMs with input-independent sparse patterns

Weaknesses:
1. While LongBench and RULER are appropriate, coverage of widely used tests is buried in the appendix and could be surfaced
2. Claims of “sub-quadratic” would be more convincing with throughput/latency/memory numbers